# Gene Regulatory Network Inference in the Presence of Selection Bias and Latent Confounders

**Gongxu Luo**[1], **Haoyue Dai**[2], **Loka Li**[1], **Chengqian Gao**[1], **Boyang Sun**[1], **Kun Zhang**[1,2]

[1] Mohamed bin Zayed University of Artificial Intelligence, [2] Carnegie Mellon University

{gongxu.luo, kun.zhang}@mbzuai.ac.ae

## Abstract

Gene regulatory network inference (GRNI) aims to discover how genes causally regulate each other from gene expression data. It is well-known that statistical dependencies in observed data do not necessarily imply causation, as spurious dependencies may arise from *latent confounders*, such as non-coding RNAs. Numerous GRNI methods have thus been proposed to address this confounding issue. However, dependencies may also result from *selection*–only cells satisfying certain survival or inclusion criteria are observed–while these selection-induced spurious dependencies are frequently overlooked in gene expression data analyses. In this work, we show that such selection is ubiquitous and, when ignored or conflated with true regulations, can lead to flawed causal interpretation and misguided intervention recommendations. To address this challenge, a fundamental question arises: can we distinguish dependencies due to regulation, confounding, and crucially, selection? We show that gene perturbations offer a simple yet effective answer: selection-induced dependencies are *symmetric* under perturbation, while those from regulation or confounding are not. Building on this motivation, we propose GISL (Gene regulatory network Inference in the presence of Selection bias and Latent confounders), a principled algorithm that leverages perturbation data to uncover both true gene regulatory relations and non-regulatory mechanisms of selection and confounding up to the equivalence class. Experiments on synthetic and real-world gene expression data demonstrate the effectiveness of our method.

## 1 Introduction

Gene regulatory network inference (GRNI [1, 2]) is fundamentally a problem of causal discovery, that is, identifying causal regulatory relationships from observational and experimental gene expression data [3, 4]. Existing GRNI studies include dependence-based methods using correlation [5, 6], regression [7, 8, 9], and mutual information [10, 11]; dynamic modeling using pseudotemporal trajectories [12, 13, 14, 15, 16]; and perturbation modeling using differential analysis [17, 18, 19, 20]. Central to these efforts is a fundamental question: can a statistical dependence observed in gene expression data be interpreted as a regulatory relationship? It is well-known that the answer is not always yes—dependencies do not imply causation, as spurious dependencies may arise from *latent confounding* such as non-coding RNAs or environmental stimuli [21, 22, 23]. Numerous GRNI methods have thus been proposed to address this confounding issue [24, 25, 26, 27].

Yet another important source of spurious dependencies remains underexplored: *selection bias*—the preferential inclusion of data points based on specific criteria [28]. In gene expression data, this arises when only cells meeting certain survival inclusion criteria are observed. As a result, two genes may appear statistically associated not because one regulates the other or they share a common regulator, but because only cells in which both genes satisfy the survival criteria persist and are sequenced. Dynamic proliferation studies support this mechanism: perturbations selectively eliminate cells that fail to meet the survival criteria, yielding the differential proliferation rates before and after perturbation [29, 30]. We show that such selection is pervasive and, if ignored or conflated with

genuine regulatory interactions, can severely bias GRNI results. Conversely, explicitly modeling selection bias can reveal non-regulatory dependencies and yield novel biological insights.

A fundamental question then arises towards GRNI under selection bias: can we distinguish dependencies due to regulation, confounding, and selection—and if so, how? This is challenging, because despite the very different biological nature between regulatory and selection processes, both of them occur upstream of the data collection process (i.e., gene screening), and may thus leave indistinguishable statistical patterns in observational data. Fortunately, with gene perturbation experiments becoming increasingly feasible in practice [31], this challenge can be effectively addressed. We show that gene perturbations offer a simple yet powerful answer to the question: selection-induced dependencies are *symmetric* under perturbation, while those from regulation or confounding are not.

To illustrate how perturbations help identify selection for GRNI, we examine a case study in leukemia cells [32]. In this dataset, the genes *AURKA* and *TOR1AIP1* exhibit strong statistical dependence that cannot be explained by any other genes, yet no known regulatory relationship between them is documented in existing databases [33]. Could this be hidden confounding? Perturbation suggests otherwise: perturbing *AURKA* produces a shift in the marginal distribution of *TOR1AIP1* expression— an outcome inconsistent with the confounding hypothesis, as perturbing a gene should not affect its upstream confounders. Furthermore, perturbing *TOR1AIP1* also leads to a notable change in *AURKA*, contradicting the asymmetric nature expected from a pure causal relationship. Together, these symmetric dependencies under perturbation point to an alternative explanation: a selection process between the two, which aligns with P53 pathway coupling analyses in cancer cells [34, 35, 36].

Building on the motivation above, we develop a flexible causal framework that models both observational and perturbation gene expression data, and allows for the presence of both latent confounders and selection bias. We characterize the information provided by perturbations through conditional independence (CI) relations in data, and show that regulatory relations, latent confounders, and selection processes typically exhibit distinct CI patterns. Based on these findings, we propose GISL (Gene regulatory network Inference in the presence of Selection bias and Latent confounders), a general nonparametric algorithm that not only identifies regulatory relations from potentially biased data, but also detects the underlying confounding and selection processes themselves, shedding lights on non-regulatory relations that, while often overlooked, also play important roles in cellular systems.

The contribution of this work is threefold. **1.** This is the first, to the best of our knowledge, to identify and to address the issue of selection bias in gene expression data and its impact on GRNI. **2**. We propose a novel algorithm for identifying regulatory relationships, as well as latent confounders and selection processes up to the equivalence class. Our algorithm is general, without requiring any parametric or graphical assumptions except for the standard ones. **3**. We validate our claims and demonstrate the effectiveness of our proposed GISL on both synthetic and real-world experimental single-cell gene expression data, showing its superiority over canonical causal discovery methods and computational GRNI baselines.

## 2 Preliminaries

### 2.1 Causal formulation of gene regulatory networks and gene perturbations

Gene regulatory networks (GRNs) represent the causal relationships governing gene activities in cells [37]. Since regulatory interactions fundamentally correspond to causal relationships, we refer to them as such throughout. We represent the whole data generating process by a Directed Acyclic Graph (DAG) $\mathcal{G}$ whose vertex set can be partitioned by $\mathcal{V} = \{\mathcal{X}, \mathcal{L}, \mathcal{S}\}$. $\mathcal{X} = \{X_i\}_{i=1}^N$ correspond to observed variables where each $X_i$ represents the expression of an individual gene. $\mathcal{L} = \{L_i\}_{i=1}^R$ accounts for the latent factors that regulate gene expression like non-coding RNAs and environmental constraints. $\mathcal{S} = \{S_i\}_{i=1}^M$ are selection variables that capture the underlying selection processes. Each $S_i$ is a binary indicator variable for an independent selection criterion, with value $1$ indicating that criterion being satisfied in a cell, and $0$ otherwise [38, 39]. Only cells with all the $S_i = 1$ are harvested in the dataset. Observed genes influenced by latent confounders and selection variables are designated as confounder pairs and selection pairs, respectively, and are formally defined as follows:

**Definition 2.1** (Confounded pair)**.** A pair $(X_i, X_j)$ is referred to as a *confounded pair*, denoted $(X_i, X_j)_l$, if and only if there exist an $L_k \in \mathcal{L}$ such that the structure $X_i \leftarrow L_k \rightarrow X_j$ exists in $\mathcal{G}$.

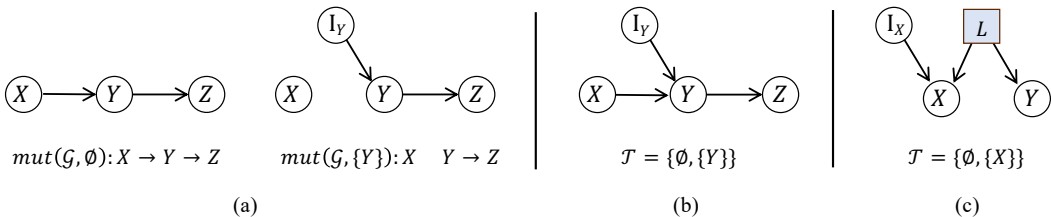

$mut(\mathcal{G}, \emptyset): X \to Y \to Z$     $mut(\mathcal{G}, \{Y\}): X \quad Y \to Z$     $\mathcal{T} = \{\emptyset, \{Y\}\}$     $\mathcal{T} = \{\emptyset, \{X\}\}$

(a)           (b)           (c)

Figure 1: Alternative graphical representations of interventions. (a) *Mutilated DAGs* depicting hard intervention [40]. (b) Generalized intervention representation using the *augmented DAG* [41]. (c) *Augmented DAG* for confounded pairs, where $L$ denotes a latent confounder [42].

**Definition 2.2** (Selection pair). A pair $(X_i, X_j)$ is referred to as a *selection pair*, denoted $(X_i, X_j)_s$, if and only if there exist an $S_k \in \mathcal{S}$ such that the structure $X_i \to S_k \leftarrow X_j$ exists in $\mathcal{G}$.

Given gene expression and perturbation data, each intervention (perturbation) target $X_k \in \mathcal{X}$ denotes variable $X_k$ is intervened on. Let $\mathcal{T} = \{T_i\}_{i=1}^{K}, T_i \subseteq \mathcal{X}$ represent the collection of intervention targets. To model the "action of do perturbation", *perturbation indicators* $\mathcal{I} = (I_i)_{i=1}^{N}$ are incorporated into the DAG as exogenous variables with directed edges pointing to corresponding intervention targets represented by an augmented DAG [43]. $I_k = 0$ indicates the observational data $D_0$, $I_k = 1$ indicates the interventional data $D_k$ with $X_k$ being intervened on. Other basic concepts are in Appendix A.

Established CRISPR-based gene perturbation methodologies encompass gene knockout (CRISPR-Cas9) and transcriptional modulation (CRISPRa/i), which can be mathematically formalized within causal inference frameworks as hard and soft interventions, respectively. For hard interventions, [40] consider each $T_k$ as factoring in a mutilated DAG over $[N]$, denoted by $mut(\mathcal{G}, X_k)$, where the edges incoming to the target $X_k$ are removed and others remain as shown in Figure 1(a).

For soft interventions, the mutilated DAG representation does not apply, as soft interventions do not remove incoming edges, and all settings may factor in the same $\mathcal{G}$. Thus, the augmented DAG is utilized as a generalized framework for representing interventions [43, 41], as illustrated in Figure 1(b), where $\mathcal{T}$ denotes the intervention target set. Intervening on a cause changes the marginal $p(\text{cause})$ and $p(\text{effect})$, but the conditional $p(\text{effect}|\text{cause})$ remains invariant. Conversely, intervening on an effect leaves $p(\text{cause})$ unchanged, $p(\text{cause}|\text{effect})$ changes [44, 45]. This invariance has been leveraged by numerous interventional causal discovery methods [46, 47, 48], predominantly implemented through parametric regression analysis. More related works are discussed in Appendix B.

## 2.2 Understanding selection bias: principles and key characteristics

Selection bias arises when only samples satisfying underlying criteria are systematically included–excluding all others–and thereby induces spurious dependencies [28]. The criteria can be categorized in two distinct ways: (1) inherent constraints ('survival'), which arise prior to treatment and are always operative, and (2) sampling bias, which results from non-randomly sampling, both of which can be conceptualized within the framework of exogenous selection in causal graphs [28, 49]. In gene expression data, inherent selection constraints predominate, notably competition for shared cellular resources [50, 51] and differential regulatory activity across promoters and enhancers [21].

**Example 1.** To illustrate how the selection process makes two independent variables statistically dependent, consider two measurements from a tumor growth study: $X$ (Concentration of inflammatory cytokine IL-6) and $Y$ (Tumor growth rate). Both variables are independent with no causal relations or confounders (both observed and latent). We simulate this independence in Figure 2(a) by sampling $X$ and $Y$ independently from a uniform distribution $\mathcal{U}[0, 2]$ (sample size $n = 3{,}000$). However,

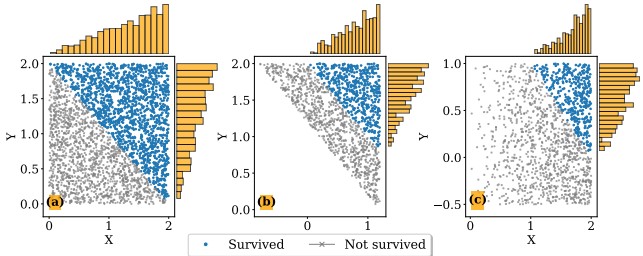

Figure 2: (a) Scatterplot of $X$ and $Y$ showing selected patients ('•') and excluded individuals ('×'). (b) and (c) Distributions after two distinct interventions on variables $X$ and $Y$, respectively, in the selected population ('•' in (a)).

the study protocol restricts analysis to patients with "favorable clinical outcomes" (i.e., those who survived and subsequently presented for hospital care). We model this selection process by retaining only cases where $X + Y > 2$, resulting in a subset of $n = 1,485$ patients (marked as • in (a)). Within this selected subset, $X$ and $Y$ appear negatively correlated–creating the illusion of statistical dependence–despite being truly independent in the full population.

To elucidate how selection processes interact with perturbations, we conducted a randomized clinical trial in which patients were assigned to either placebo or treatment arms. The control group follows distribution $P^c$, which is the '•' in (a). For the treatment group, we implement soft deterministic interventions on $X$, specifically $do(X = X - 0.5)$. The resulting distribution $P^{t_x}$ (post-intervention) is visualized by '•' in (b) (n=514). When intervention interacts with selection criteria, we observe that $P^{t_x}(Y) \neq P^c(Y)$, despite the absence of causal relations between $X$ and $Y$. This discrepancy arises because the consistent application of selection constraints filters out specific samples, altering the distribution of $Y$. Similarly, under a hard stochastic intervention on $Y$, $do(Y \sim \mathcal{U}[-0.5, 1])$, the distribution $P^{t_y}$ is represented by '•' in (c) (n=386). We observe that $P^{t_y}(X) \neq P^c(X)$. The distribution changes induced by selection after different interventions on both sides, along with the corresponding changes in sample size $n$, illustrate the symmetry perturbation effect of selection processes and its underlying mechanisms through **criterion-based filtering**.

With all the notions ready, the joint probability density of intervention over $\mathcal{X}$ is as follows:

$$p^{T_k}(\mathcal{X}) = f_s(\mathcal{X}) \prod_{\{i|X_i \in T_k\}} p^{T_k}(X_i|X_{pa_{\mathcal{G}}(i)}) \prod_{\{j|X_j \notin T_k\}} p^{\emptyset}(X_j|X_{pa_{\mathcal{G}}(j)}), \tag{1}$$

where $p^{\emptyset}$ and $p^{T_k}$ denotes the probability density of observational distributions and interventional distributions with intervention target $T_k \subset \mathcal{X}$, $f_s(\mathcal{X})$ indicates the selection constraints on observational sets $\mathcal{X}$, and $pa_{\mathcal{G}}(i)$ indicates the parents of $X_i$ in $\mathcal{G}$ [52]. The joint probability density of observation is $p^{T_k}(\mathcal{X}), T_k = \emptyset$. Note that $p^{T_k}(X_j|X_{pa_{\mathcal{G}}(j)}) = p^{\emptyset}(X_j|X_{pa_{\mathcal{G}}(j)}), \forall X_j \notin T_k$.

## 3 Methodology

Having laid out the necessary preliminaries, we now turn to interpreting the observed statistical dependencies. We first establish the rationale for leveraging gene perturbation data to detect selection bias, uncover latent confounders, and infer regulatory relations (§ 3.1). Subsequently, we present a computational framework and algorithmic solutions tailored to address these challenges (§ 3.2).

### 3.1 Differentiating causal relations, selection processes, and latent confounders

We now describe how causal relations, selection processes, and latent confounders can be distinguished via perturbations, focusing on two key patterns: symmetry and perturbation effects.

**Perturbation symmetry.** Causal relationships ($X \rightarrow Y$) are asymmetric: perturbing $X$ shifts the distribution of $Y$, but not vice versa (Figure 3(a)). Selection processes ($X \rightarrow S \leftarrow Y$) are symmetric: perturbing either shifts the other, as discussed in Example 1 in Section 2.2 (Figure 3(c)). In the case of latent confounders, unlike selection, perturbations do not propagate through the latent confounder $L$, *i.e.*, perturbing $X$ or $Y$ does not affect the other (Figure 3(b)). These contrasts provide a basis for differentiating causal, selection, and confounding structures. Note that throughout we focus on acyclic graphs, as our primary goal is to provide a proof of concept for addressing selection bias. Cyclic structures introduce additional complexities not essential for this purpose. Once the acyclic case is fully understood, extending the approach to handle cycles becomes more straightforward.

**Perturbation effects.** However, perturbation symmetry alone is insufficient when multiple dependencies coexist. For instance, both pure regulatory relationships (Figure 3(a)) and regulation involving latent confounders (Figure 3(d)) are perturbation asymmetric, making them indistinguishable. In terms of *perturbation effects*, fortunately, by modeling observational and perturbation data within the augmented DAG framework ($\mathcal{V} = \{\mathcal{X}, \mathcal{L}, \mathcal{S}, \mathcal{I}\}$), where the perturbation indicators $\mathcal{I}$ are incorporated as exogenous variables, we can model how perturbation leads to changes in distribution on other genes via capturing the dependencies between $\mathcal{I}$ and affected ones. To see the power of this new framework, referring back to structures (d) and (a), since conditioning on $X$ constrains the distribution of $Y$ via $L$, conditional distribution changes between $P(Y|X, I_X = 1)$ and $P(Y|X, I_X = 0)$ will be

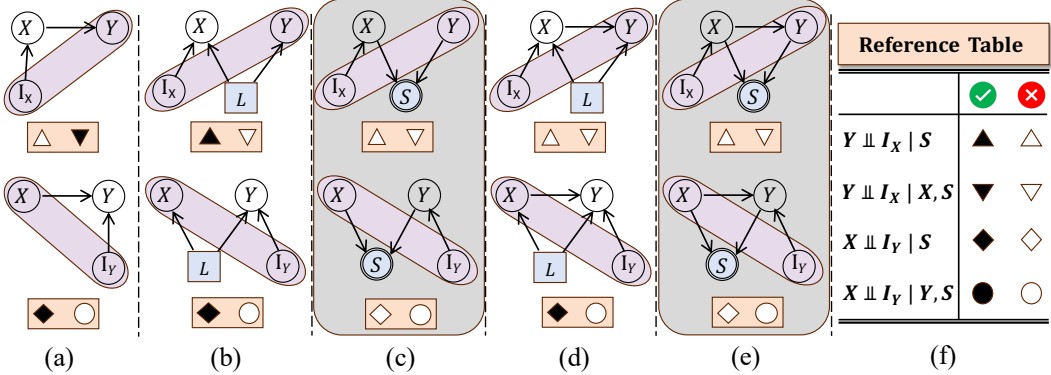

Figure 3: Distinguishing causal, selection, and confounding structures via perturbation effect and symmetry. ▨ indicates the targeted gene pairs of CI test. (a) refers to the direct cause structure between X and Y (represented by 'C'). (b) means there is a latent confounder between them ('L'). (c) is the structure of selection process ('S'). (d) stands for causation and latent confounders at the same time ('C & L'). (e) stands for causation and selection process at the same time ('C & S'). ▨ contains CI results. (f) serves as a reference table summarizing the CI patterns for each target gene pair: different symbols correspond to different CI relations; black symbols (▲, ▼, ♦, ●) indicate the conditional independence, while white symbols (△, ▽, ◊, ○) indicate the conditional dependence. For example, (a) encodes four CI relations: $Y \not\perp\!\!\!\perp I_X \mid S$ (△) and $Y \perp\!\!\!\perp I_X \mid X, S$ (▼) at the top; $X \perp\!\!\!\perp I_Y \mid S$ (♦) and $X \not\perp\!\!\!\perp I_Y \mid Y, S$ (○) at the bottom.

observed in (d), resulting in $I_X \not\perp\!\!\!\perp Y \mid X$. In contrast, structure (a) lacks this conditional dependence, making the two structures distinguishable. This complementary use of perturbation effects alongside symmetry enables us to distinguish different structures.

Having outlined the distinct perturbation symmetries and perturbation effects associated with the three causal structures, we are now able to categorize the observed dependencies accordingly. To achieve this, we apply conditional independence (CI) tests—powerful statistical tools capable of detecting changes in distributions between the interested variables. Their application relies on two mild assumptions, a special case is further discussed in 3.4, as follows:

**Assumption 3.1** (Causal Markov assumption). Given a DAG $\mathcal{G}$ over the variable set $\mathcal{V}$, every variable $M$ in $\mathcal{V}$ is probabilistically independent of its non-descendants given its parents in $\mathcal{G}$.

**Assumption 3.2** (Faithfulness assumption). Given a DAG $\mathcal{G}$ and distribution $\mathcal{P}$ over $\mathcal{V}$, $\mathcal{P}$ implies no CI relations not already entailed by the Markov assumption.

*Remark* 3.3. These two assumptions are common and fundamental for connecting causality with statistical tools, as they ensure that CI tests can correctly capture the underlying causal structure.

Under these mild assumptions, we can now reliably employ the CI test tool to capture distribution changes manifested by perturbation symmetry and effects. The resulting CI patterns between $\mathcal{I}$ and $\mathcal{X}$ exhibit distinct behaviors across different structural scenarios, as illustrated in Figure 3 ▨. Accordingly, causal relationships, selection processes, and latent confounders are distinguishable.

*Remark* 3.4. When selection interacts with CI testing, we observe extra conditional dependence between indicator $I$ and variables in selection pairs, such as $I_X \not\perp\!\!\!\perp Y \mid X$ (▽), and $I_Y \not\perp\!\!\!\perp X \mid Y$ (○), as illustrated in the gray-shaded columns (c) and (e) of Figure 3 **only when the variable is both intervened and is in the conditional set**. This is because testing the dependence between $I_X$ and $Y$ given $X$ is equivalent to detecting differences between the distributions $P(Y|X, I_X = 0)$ and $P(Y|X, I_X = 1)$. Referring to the pre-intervention case (a) and post-intervention case (c) in Figure 2, it is obvious that $P(Y|X, I_X = 0) \neq P(Y|X, I_X = 1)$, which establishes that $I_X \not\perp\!\!\!\perp Y|X$. Although this special case violates both Markov and faithfulness assumptions, the distinctive characteristic provides an opportunity to identify selection. The observed four unique CI patterns are shown in Figure 3, i.e., (a), (b), (c), (d).

## 3.2 Algorithm GISL: Handling both selection bias and latent confounding

Building on the different CI patterns across distinct causal structures in Section 3.1, we develop Algorithm 1, named Gene Regulatory Network Inference in the presence of Selection bias and

**Algorithm 1:** Gene Regulatory Network Inference in the presence of Selection bias and Latent confounders (GISL).

---

**Input:** Observational $D_0$ and single gene perturbation data $\{D_k \mid X_k \in \mathcal{T}\}$ over $\mathcal{X}_{[N]}$.
**Output:** Equivalence class $\mathcal{G}$ over $\mathcal{X}$, Confounder pairs $\boldsymbol{L}$, Selection pairs $\boldsymbol{S}$
**Step 1. Initialize:** Set $\mathcal{G}^0$ as a fully undirected graph; $\boldsymbol{L} = \{\ \}, \boldsymbol{S} = \{\ \}, \boldsymbol{L}' = \{\ \}$, Unk $= \{\ \}$.
**Step 2. Recover the skeleton from observational data $D_0$:** $\mathcal{G}^1 \leftarrow FGES(\mathcal{G}^0, D_0)$.
**Step 3. Capture CI patterns from observational data and perturbation data:**
> **foreach** $(X_i, X_j) \in \mathcal{G}^1$ and $(X_i, X_j) \subseteq \mathcal{T}$ **do**
>> Compute $\mathcal{J} = \{[I_i, X_j \mid \mathcal{S}], [(I_i, X_j) \mid X_i, \mathcal{S}], [I_j, X_i \mid \mathcal{S}], [(I_j, X_i) \mid X_j, \mathcal{S}]\}$
>> **If** $\mathcal{J} ==$ Figure 3(a) **then** $\mathcal{G}^2 \leftarrow \mathcal{G}^1$.            $\Leftarrow$*Regulatory Relationship*
>> **If** $\mathcal{J} ==$ (b) **then** $\boldsymbol{L} \leftarrow (X_i, X_j)$.                $\Leftarrow$*Latent Confounders*
>> **If** $\mathcal{J} ==$ (c) **then** $\boldsymbol{S} \leftarrow (X_i, X_j), \mathcal{G}^2 \leftarrow \mathcal{G}^1$.      $\Leftarrow$ *Selection Bias*
>> **If** $\mathcal{J} ==$ (d) **then** $\boldsymbol{L}' \leftarrow (X_i, X_j)$.                  $\Leftarrow$'C & L'
>> **If** $\mathcal{J} ==$ other patterns not included in Figure 3, **then** Unk $\leftarrow (X_i, X_j)$

**Step 4. Correct CI patterns by blocking d-separated paths:**
> **repeat** Step 3 $(X_i, X_j) \in \boldsymbol{L}', \boldsymbol{S}$, Unk, conditioning on the subsets of their non-endpoints.
>> **If** $(X_i, X_j) \in \boldsymbol{L}'$ and correct to (a) **then** $\boldsymbol{L}' := \boldsymbol{L}' \setminus \{(X_i, X_j)\}$
>> **If** $(X_i, X_j) \in \boldsymbol{S}$ and correct to (a) or (b) or (c) **then** $\boldsymbol{S} := \boldsymbol{S} \setminus \{(X_i, X_j)\}$
> **until** *no further edges* $(X_i, X_j)$ *need correcting*

**return** $\mathcal{G}^2, \boldsymbol{L} \cup \boldsymbol{L}', \boldsymbol{S}$.

---

Latent confounders (GISL), a general nonparametric algorithm, to detect the existence of selection bias and latent confounders, and identify regulatory relationships. We first obtain adjacencies from observational data $D_0$, as it provides the sparest skeleton by statistical criteria such as CI tests or the Bayesian Information Criterion (BIC) [53]. The resulting skeleton, representing conditional dependencies, guides further exploration of underlying causal structures (**Step 2**).

To capture the differences in symmetry and perturbation effects, we examine CI patterns between perturbation indicators $\mathcal{I}$ and $\mathcal{X}$ from both observational data $D_0$ and single gene perturbation data $\{D_k \mid X_k \in \mathcal{T}\}$ to identify structures that can inform the skeleton as illustrated in Figure 3(f) for gene pair $(X, Y)$. The CI results are collected and represented by the set $\mathcal{J}$ in **Step 3**. While examples offer initial intuition in distinguishing different causal structures, more complex scenarios require further careful analysis for completeness, especially when multiple paths connect node pairs, including d-separated paths (representing conditional independence) and inducing paths (representing conditional dependence), defined as follows:

**Definition 3.5** (d-separation [54])**.** Let $\mathcal{G}$ be a DAG, and let $\mathcal{A}$, $\mathcal{B}$, and $\mathcal{C}$ be three disjoint sets of nodes in $\mathcal{G}$. We say that $\mathcal{A}$ and $\mathcal{B}$ are d-separated by $\mathcal{C}$ in $\mathcal{G}$ if and only if every path between a node in $\mathcal{A}$ and a node in $\mathcal{B}$ is blocked by $\mathcal{C}$.

**Definition 3.6** (Inducing path [55])**.** In a DAG with $\mathcal{L}$ and $\mathcal{S}$, $X, Y$ are any two vertices, and $\mathcal{L}, \mathcal{S}$ are disjoint sets of vertices not containing $X, Y$. A path $p$ between $X$ and $Y$ is called an inducing path relative to $\langle \mathcal{L}, \mathcal{S} \rangle$ if and only if every non-endpoint vertex on $p$ is either in $\mathcal{L}$ or a collider, and every collider on $p$ is an ancestor of either $X, Y$, or a member of $\mathcal{S}$.

If the inferred structure includes both inducing paths and d-separated paths, further correction is required to remove the dependencies induced by d-separated paths (**Step 4**), as illustrated in Example 2 in Section 3.2. More complex inducing-path configurations and their implications for recovering the true causal structure are discussed in Section C. Given the adjacencies provided by the skeleton, d-separated paths can be blocked by conditioning on adjacent nodes. After correction, GISL can detect the presence of selection bias and latent confounders between each pair of genes as well as regulatory relationships. To represent equivalence classes under latent confounding and selection bias, we adopt the edge semantics from the Partial Ancestral Graph (PAG) framework.

**Example 2** Consider a simple case involving two variables $X$ and $Y$, connected by two paths: an inducing path $X \leftarrow L \rightarrow Y$, and a d-separated path $X \rightarrow Z \rightarrow Y$. Without conditioning on $Z$, the observed CI pattern reflects the influence of both paths—specifically, both confounding via $L$ and dependency via $Z$, denoted as "C & L". Only by conditioning on Z can we block the d-separated path and correctly identify the underlying confounding structure, yielding the CI pattern "L".

Table 1: Accuracy % of GISL in identifying selection bias on synthetic data. We report the mean and variance values of accuracy across 10 independent graphs for each configuration.

| $n$ \ $|\mathcal{X}|$ | 10 | 15 | 20 | 25 | 10 | 15 | 20 | 25 |
|---|---|---|---|---|---|---|---|---|
| | | Hard intervention | | | | Soft intervention | | |
| **500** | $63.3\pm22.7$ | $68.7\pm20.2$ | $72.0\pm22.2$ | $70.0\pm20.0$ | $60.4\pm24.0$ | $60.6\pm16.6$ | $68.2\pm17.3$ | $67.5\pm15.9$ |
| **1,000** | $70.4\pm20.0$ | $70.0\pm19.6$ | $74.6\pm18.6$ | $75.9\pm14.9$ | $75.0\pm16.7$ | $74.8\pm20.2$ | $80.2\pm0.7$ | $72.1\pm14.2$ |
| **1,500** | $71.0\pm22.3$ | $72.5\pm18.6$ | $77.5\pm17.5$ | $76.9\pm17.2$ | $72.5\pm15.6$ | $80.8\pm11.3$ | $78.2\pm13.6$ | $77.5\pm16.3$ |
| **2,000** | $73.4\pm22.6$ | $75.4\pm19.6$ | $75.7\pm15.4$ | $73.9\pm14.2$ | $80.0\pm11.0$ | $73.3\pm17.3$ | $78.2\pm13.6$ | $75.1\pm14.7$ |

## 3.3 Identifiability result of the GISL algorithm

We analyze the identifiability of the proposed GISL framework in detecting selection bias, latent confounders, and regulatory relationships and conclude the following theorem:

**Theorem 3.7.** *(Identifiability of GISL)* *Let the observational and perturbation data be generated from the DAG model $\mathcal{G}$ defined in Equation* (1). *Under Markov 3.1 and faithfulness 3.2 assumptions, when the sample size $n \to \infty$, the causal relationships, selection processes, and latent confounders are identifiable up to the equivalence classes of four types of CI patterns in Figure 3 among variables that are subject to interventions. Moreover, the presence of selection processes and latent confounders (existing or not) is identifiable.*

*Remark* 3.8. Identifiability can be established intuitively. Take the causal relation $X \to Y$ as an example. Structurally, $X$ can only point out with tails, and $Y$ can only be pointed to with arrowheads. To reproduce this structure using other inducing paths, one must substitute the tail with a v-structure ($\mathcal{S}$) or the arrowhead with the hidden common cause ($\mathcal{L}$). If selection is used to replace the tail, then $I_X$ and $Y$ would not be conditionally independent given $X$ (discussed in 3.4), which contradicts the CI pattern of 'C' in Figure 3(a). Alternatively, if a latent confounder replaces the arrowhead, since hidden common causes only contribute to two arrowheads, the requirement of tails and removing the effect of the extra arrowhead necessitates an intermediate node to keep the CI patterns with 'C'. As the intermediate node cannot be blocked, this requires a v-structure and an edge point to $Y$ following Definition 3.6. Then, $X$ must point to the intermediate node to form the V-structure with the extra arrowhead. Therefore, $X$ is the ancestor of $Y$. For more details on proofs of the identifiability of GISL in the presence of latent confounders and selection process, please kindly refer to Appendix D.

## 4 Experiments

We begin by evaluating GISL's ability to detect selection bias on synthetic data. Next, we present the benefits of considering selection bias in causal discovery, specifically, benchmark its performance in identifying regulatory relationships tasks, against established baselines. Finally, we apply GISL to real-world gene expression data, using Z-scores as a proxy ground truth.

### 4.1 Identify the selection bias on synthetic data

**Nonparametric settings.** To better reflect the complexity of gene expression, we adopt a nonparametric structural causal model (SCM) that accommodates latent confounders and selection bias without assuming specific functional or distributional forms: $L_k = E_{l_k}, X_i = (1 - I_i)\big[f_\emptyset\big(\text{pa}_{\mathcal{G}}(X_i)\big) + E_{x_i}\big] + I_i\big[f_i\big(\text{pa}_{\mathcal{G}}(X_i)\big) + E_{x_i}\big]$, with selection governed by $f_s(X_i, X_j) > C$. $E_{x_i}$ and $E_{l_k}$ are Gaussian noise terms with randomly selected means and variances, $C$ is randomly selected threshold, and $f_\emptyset, f_s$ are sampled from diverse nonlinear functions: linear, square, sin and log. $I_i \in [0, 1]$ indicates the gene is perturbed or not, with corresponding perturbation function $f_i$. Following Section 2.1, we simulate gene perturbations using both hard and soft interventions. In an ideal hard intervention (knockout), one would set $do(X_i = 0)$. But to capture off-target variability, we instead sample $do(X_i \sim \mathcal{U}(a, b))$, with the interval $[a, b]$ randomly chosen for each intervention. Soft interventions (knockup and knockdown) model up- or down-regulation by adding a uniform noise to the original expression: $X_i \leftarrow X_i + \varepsilon$, where $\varepsilon \sim \mathcal{U}(0, c)$ for knockup, and $\varepsilon \sim \mathcal{U}(-d, 0)$ for knockdown. The magnitudes $c$ and $d$ define the strength of the up- or down-regulation.

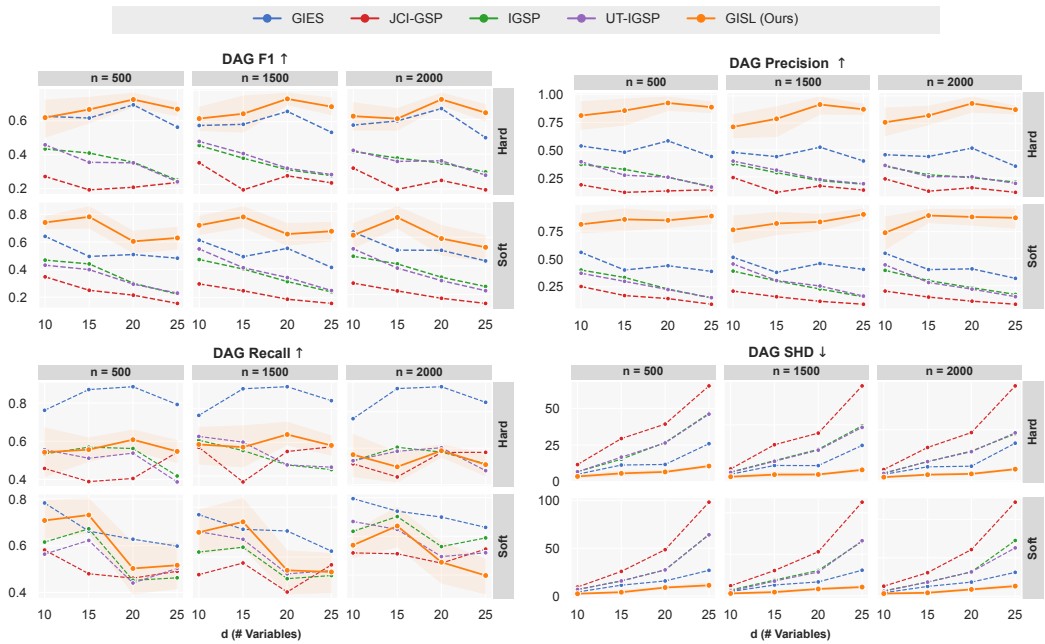

Figure 4: Comparison results in identifying regulatory relations under four metrics: DAG $F_1$, DAG Precision, DAG Recall, and DAG SHD (Structural Hamming Distance). All values are averaged over 10 runs with different random seeds. Error bars represent the 95% confidence interval.

**Synthetic data generation.** We first instantiate DAGs using Erdős–Rényi model [56] with the number of edges equal to the number of variables. Then groups of synthetic gene expression data are generated following the SCM and intervention protocols detailed above, sweeping the configurations: the number of observation data $n$ ranged in $\{500, 1500, 2000\}$, the number of variables $|\mathcal{X}| \in \{10, 15, 20, 25\}$. For each $(n, |\mathcal{X}|)$ configuration, and separately for hard and soft intervention, we instantiate 10 independent structures by randomly choosing 1-3 selection pairs and 1-3 confounding pairs, reflecting the complexity of the gene expression scenario.

**Comparison results (selection bias, Table 1 and Table 2).** We report the evaluation results of GISL (KCI for CI test with $\alpha = 0.05$) in identifying selection bias on the synthetic data detailed above. The accuracy score is defined as the percentage of identified selection pairs that are aligned with ground truth in all positive predictions. **In Table 1**, GISL accurately captures selection bias, especially when more data points are collected. Take an example with $|\mathcal{X}| = 20$, leveraging more data to precisely characterize distribution changes yields a $\sim$7 % accuracy gain. Furthermore, to evaluate GISL under increasing selection complexity, we fix $|\mathcal{X}| = 15$ and $n = 2000$,

Table 2: Accuracy % under more selection pairs; Upper-bound baseline: Collider identification.

| $|f_s|$ | 1 | 2 | 3 | 4 |
|---|---|---|---|---|
| Hard | $82.8 \pm 14.8$ | $81.6 \pm 11.2$ | $68.1 \pm 18.1$ | $70.8 \pm 20.6$ |
| Soft | $78.5 \pm 15.6$ | $76.1 \pm 14.5$ | $70.6 \pm 17.7$ | $69.5 \pm 14.3$ |
| Collider | $90.0 \pm 5.0$ | $85.0 \pm 5.3$ | $83.3 \pm 5.6$ | $82.5 \pm 3.9$ |

and sweep the number of selection processes ($|f_s| \in \{1, 2, 3, 4\}$) **in Table 2**, alongside the identification of collider as an approximate upper bound. As more genes are subjected to biased selection, the induced observational constraints make detection increasingly challenging for CI-based methods. Despite this, GISL maintains strong performance across all settings, detecting selection bias in around 70% of affected gene pairs—even when more than half the variables are biased. To the best of our knowledge, no existing methods are designed to identify selection bias arising from survival constraints in gene regulatory tasks. As such, we do not report baseline comparisons in this section.

## 4.2 Identify regulatory relationships on synthetic data

**Baselines.** To rigorously evaluate GISL's ability to disentangle true regulatory relationships from spurious dependencies–particularly those arising from selection bias–we conduct experiments on synthetic data, comparing GISL against robust baseline methods: GIES [40], IGSP [17], UT-IGSP [57] and JCI-GSP method used in [57], which is an extension of JCI [58] with GSP [59].

**Benchmarking results (regulatory relationships). Figure 4** illustrates that GISL achieves superior overall performance in identifying regulatory relationships (DAG $F_1$). Among evaluation metrics, precision emerges as most critical, as it directly reflects GISL's advantage in disentangling regulatory relationships from spurious dependencies induced by selection bias. While existing approaches often misattribute selection biases as causal relationship or confounders, GISL effectively distinguishes these phenomena, resulting in markedly improved precision. We emphasize that the objective of this paper is not to propose a definitive solution to causal discovery, but rather to highlight that ignoring selection bias can lead to flawed casual interpretation. Further comparisons including with FCI [55], ICD [60], and non-causal baselines are presented in Appendix E. Moreover, the evaluation in the robustness of GISL across noise levels, structural and functional complexity of the SCM, and larger graphs are discussed in Appendix E.

## 4.3 The presence of selection bias on single-cell gene expression data

**Datasets.** We use three real-world scRNA-seq perturbation datasets: Dixit [32] and Adamson [61], from K562 leukemia cells (5,012 and 5,060 genes under 19 and 86 single-gene perturbations), and Norman [31], from A549 lung carcinoma cells (5,045 genes under 105 perturbations). Each perturbation targets one gene and is profiled across numerous cells.

**Selection bias identification on real-world data (Figure 5).** We plot the detection result on the Dixit dataset in Figure 5. Results for the other two datasets and detailed analysis are attached in Appendix F.1. Our method can detect both the underlying selection processes and latent confounders at the same time, facilitating the explanation of observed dependence among genes (regulatory relationship, selection bias, and latent confounders). In practice, the explanation of the selection process and latent confounders will guide the biologists that perturbing these genes may not have the expected effect, as the selection process may lead to unaffordable consequences after sample filtering, and latent confounders will have no reaction to perturbation.

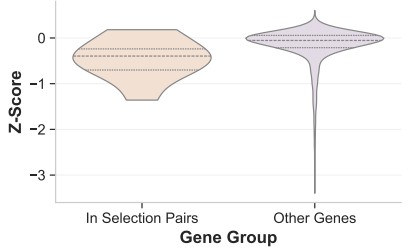

Figure 5: Experimental result on 19 perturbed key genes from perturb-seq [32]. **S** and **L** imply the detected selection and confounded pairs, blue edges are the regulatory interactions priorly known [33].

**Evaluation using Z-score.** To evaluate whether the identified gene pairs are truly subject to selection bias–without relying on ground-truth labels–we introduce Z-score: a score function that captures the changes in growth rate before and after perturbations [29], detailed calculation refers to Appendix G. Notably, both the increase (+) and decrease (-) result in a Z-score absolute value larger than $0$. We have the following assertion: *true selection pairs must exhibit high absolute Z-score values, while the opposite is not necessarily true*. To see this, consider $X_i \rightarrow S \leftarrow X_j$, where $X_i$ and $X_j$ are genes under the selection process $f_s$. Perturbing either gene causes differential cell survival and, thus, a higher Z-score. However,

Figure 6: Comparison of Z-score distribution between genes in selection pairs and others.

a high Z-score does not necessarily imply direct selection processes: in $X_i \rightarrow S \leftarrow X_j \leftarrow X_k$, perturbing $X_k$ indirectly alters the joint distribution of $(X_i, X_j)$ under $S$, yielding a high Z-score for $X_k$ even though $X_k$ is not under direct selection process. Therefore, only gene pairs for which both exhibit high Z-scores can be selection pairs, and simultaneously, an accurate algorithm for detecting selection pairs should also perform well on detecting pairs with high Z-scores. By applying a Z-score threshold of 0.15 to define high-scoring gene pairs, GISL achieves accuracies of **77.7**% on Dixit (see Figure 6 for comparison in Z-score distribution), **80.2**% on Adamson, and **74.8**% on Norman.

# 5    Conclusion and Discussion

We introduce a novel perspective to gene regulatory network inference (GRNI): selection bias, particularly the often-overlooked survival constraints, in shaping dependencies. Building on theoretical insights into the distinct perturbation symmetries and effects associated with regulation, confounding, and selection, we propose GISL—a general nonparametric algorithm capable of disentangling regulatory relationships, latent confounders, and, crucially, selection-induced dependencies.

Empirical analyses on large-scale gene expression datasets reveal pervasive selection bias in real-world scenarios. Extensive benchmarking on synthetic data further demonstrates that (1) failure to account for selection bias undermines the validity of GRNI methods, and (2) explicitly modeling such bias, as done by GISL, yields more accurate and robust causal discovery. Limitations and broader implications of this work are discussed in Appendix H.

## Acknowledgment

We would like to acknowledge the support from NSF Award No. 2229881, AI Institute for Societal Decision Making (AI-SDM), the National Institutes of Health (NIH) under Contract R01HL159805, and grants from Quris AI, Florin Court Capital, and MBZUAI-WIS Joint Program, and the Al Deira Causal Education project.

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

# Appendix

## A  Concepts

**Definition A.1** (Marginal independence test). Check whether two variables $X$ and $Y$ are independent of each other without considering any other variables. Mathematically: $X \perp\!\!\!\perp Y$, if and only if X and Y are independent in the overall data distribution.

**Definition A.2** (Conditional independence test). Evaluate whether two variables $X$ and $Y$ are independent given a third variable or set of variables $Z$. Mathematically: $X \perp\!\!\!\perp Y | Z$, if and only if $X$ and $Y$ are independent conditioned on $Z$.

**Definition A.3** (d-connection). Every path from a node in $X$ to a node in $Y$ is d-connected by $Z$, if and only if $X$ and $Y$ are always conditionally dependent given $Z$.

**Definition A.4** (Partial identifiability). The causal graph is partially identified if and only if not all causal structures can be uniquely determined from the available data and assumptions, and only a set of plausible structures can be uniquely determined.

**Definition A.5** (Identifiability). The causal graph is identified if and only if all causal structures are uniquely determined from the observational data.

## B  Related work

In this section, we provide a comprehensive review of the literature on causal discovery methods and briefly categorize other computational Gene Regulatory Network Inference (GRNI) methods.

### B.1  Gene regulatory network inference

Over the past decades, numerous methods for GRNI have been developed, encompassing computational and causal approaches. Computational models, represented by a boolean model, differential equation, gene correlation, and correlation ensemble over pseudo-time, focus on exploring dependencies among genes [62, 13, 63, 64, 65, 66]. These methods focus on dependencies without explicitly considering causal relationships. In contrast, causal models go beyond mere dependence, aiming to uncover genuine causal relationships within Gene Regulatory Networks (GRNs) [17, 18, 67].

### B.2  Causal discovery

There are constraint-based causal discovery [3, 68], score-based ones [69, 70], and methods that utilize properties of functional forms in the true causal process [71, 72, 73]. There is subset of work considering causal discovery as a continuous optimization problem [74]. These pioneering works provide comprehensive frameworks that follow different principles for causal discovery. However, they have not yet addressed the challenges posed by latent confounders and selection bias.

**Causal discovery under latent confounders.** When latent confounders are present, confounding effects give rise to spurious dependencies, complicating causal discovery. The FCI [75, 55] algorithm was the first to address this challenge, producing equivalence classes represented by Partial Ancestral Graphs (PAGs). Building on FCI's results, [76] leveraged the Triad condition to identify shared latent confounders and infer causal relationships between measured variables under linear constraints. [77] utilized an autoencoder framework to reconstruct nonlinear causal relationships among observed variables while simultaneously accounting for the presence of latent confounders. [78] estimated the mixing matrix using higher-order cumulants and introduced the testable One-Latent-Component condition to identify latent variables and establish causal orders. In the context of GRNI, some studies have begun addressing the issue of latent confounders, with a focus on recovering these confounders [79] and uncovering causal relationships among genes [19].

**Causal discovery under selection bias.** Some fundamental works focused on identifying selection and causal relationships by finding Y-structure [80], identifying selection bias under certain parametric assumptions [81], studying the identifiability and estimation of functional causal models under the outcome-dependent selection structure [82], and recovering the conditional probability from selection-biased data [83]. However, these methods are limited to either parametric assumptions, i.e., linear

Gaussian, or outcome-dependent selection structure, which are unsuitable for the general setting (nonparametric, latent confounders, selection bias) and the pairwise selection context of GRNI.

**Causal discovery under latent confounders and selection bias** When latent confounders and selection bias are present, their hidden nature introduces spurious dependencies between observed variables. These dependencies compromise the core properties used in causal discovery, leading to a loss of identifiability and making it difficult to distinguish true causal relationships. The FCI algorithm [75, 55] was the first attempt to discover ancestral relationships, but it is limited to identifying an equivalence class constrained by the structural information of v-structures. Additionally, significant ambiguities persist regarding the selection structure. Similarly, other approaches have resulted in ancestral equivalence classes that are constrained by graphical properties [60].

### B.3  Interventional Causal discovery

**Early Attempts in Interventional Causal Discovery.**  The earliest Bayesian approaches [84, 85] estimated the posterior distribution of DAGs using both observational and interventional data. However, these methods did not address key challenges such as identifiability or equivalence class characterization. [86] was the first to explore identifiability and Markov equivalence of interventional causal discovery. They focused on single-variable interventions with mechanism changes (soft interventions) and provided a graphical criterion for determining when two DAGs are indistinguishable, though no formal representation of the resulting equivalence class was introduced.

**When latent confounders are involved.**  In the pure observational data and nonparametric causal discovery setting, the frameworks of MAG and FCI have been well established [87, 55]. For interventional causal discovery, various methods have been proposed to address latent variables based on measuring overlapping variables across different interventions [88, 46, 89] and invariance [90, 85, 42]. They are either lying under the umbrellas of FCI and the augmented DAG frameworks or using parametric assumptions.

**When selection is involved**  In purely observational and nonparametric causal discovery, selection is typically constrained by structural limitations [82]. However, in the context of interventional causal discovery, various methods have been developed to explicitly address selection bias. These methods leverage interventional data to disentangle the effects of selection mechanisms from genuine causal relations [91, 58]. Similarly, these methods are still limited to equivalence classes under the umbrella of FCI. Moreover, [92] discussed how selection interacts with intervention, and built a twin interventional graph to model the selection that happens before intervention.

## C  Understanding inducing paths

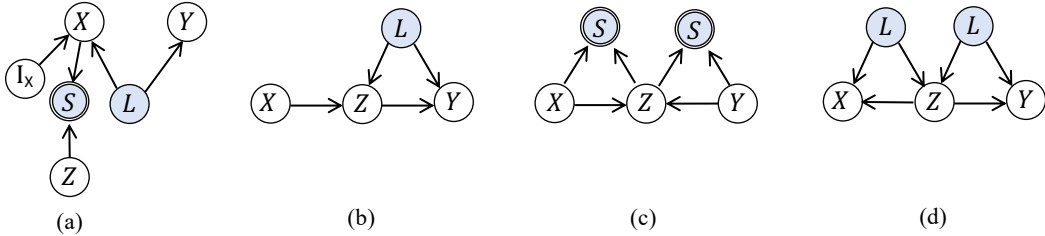

(a)  (b)  (c)  (d)

Figure 7: Illustrations of inducing paths, where $X$ and $Y$ are always d-connected.

A most generalized model might include latent confounders, perturbation indicators, and observed variables involved in the selection process. Nonetheless, such generalized assumptions often render causal relationships indeterminate, making the results less informative.

For example, latent confounders can still make the direct causal relations unidentifiable. Consider the case $X \to Z \to Y$ with a latent confounder $L$ pointing to both $Z$ and $Y$ shown in Figure 7 (b). Adding a direct edge $X \to Y$ renders the scenarios equivalent, even if perturbation data $I_X, I_Y$ are available, as the dependence between $X$ and $Y$ cannot solely be explained by $Z$. With the selection process, in the model $S \leftarrow X \leftarrow L \to Y$ (Figure 7 (a)), whether or not to add a direct edge $X \to Y$, the two scenarios are unidentifiable. The causal relationship is identified as limited to ancestor

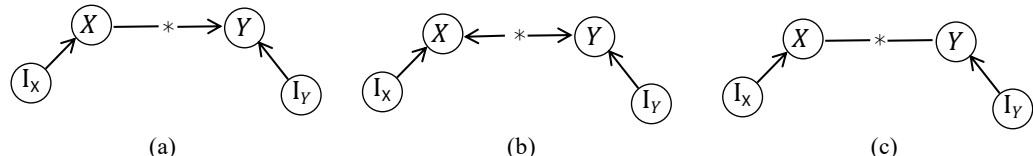

| (a) | (b) | (c) |

Figure 8: Required structure for causal relation, latent confounders, and selection process, where $*$ means the nodes on the paths cannot be blocked.

relationships or inducing paths. Similarly, Figure 7 (c) cannot be distinguished from $X \to S \leftarrow Y$, and Figure 7 (d) cannot be distinguished from $X \leftarrow L \to Y$.

After understanding inducing paths, these paths (which are always d-connected) can mimic any structure reflected in conditional independence (CI) patterns. Consequently, the identifiability of causal relationships, latent confounders, and the selection process remains constrained by the true structure or the inducing path. In the representation of ancestral graphs, they are identifiable up to equivalence classes represented by PAG. The details of the proof can be found in the Appendix D.

## D Proof

**Latent confounders:** The unique structure involving latent confounders is represented by the collider configuration $I_X \to X \leftarrow L \to Y \leftarrow I_Y$. It needs both $X$ and $Y$ to be pointed to with an arrowhead shown in Figure 8(b). Any nodes with tails will corrupt the conditional dependence between $I_X$ and $Y$, and $I_Y$ and $X$. Therefore, causal relations and selection processes cannot be directly involved. Only $\leftarrow\!\!\to$ is allowed, resulting in d-separated paths between $X$ and $Y$. Then, the latent confounders can be identified, as the CI pattern remains unaffected when these d-separated paths are blocked. Only when cases with inducing paths, such as the scenario depicted in Figure 7(d), cannot be identified. This is because the d-connected paths between $X$ and $Y$ mimic the same unique structures associated with latent confounders. However, this d-connection must have latent confounders to provide the arrowhead, leading to the identifiability of the existence of latent confounders.

**Selection process:** The unique structure of the selection process, which needs both $X$ and $Y$ only points out with tails, which are characterized by the paths $I_X \to X \to S$ and $I_Y \to Y \to S$. As discussed in Remark 3.4, this structure exhibits both dependence and conditional dependence between $I_X$ and $Y$, and between $I_Y$ and $X$. These conditional dependencies need both tails and arrowheads to interact together. However, without selection processes providing vstructure and d-connection, only causal relations and latent confounders will never meet the requirements of interaction of tails and arrowheads for $X$, $Y$, and middle nodes $*$ that can not block the path. Only cases involving selection processes forming the inducing path shown in Figure 8(c) were satisfied. Although there is no direct selection work on $X$ and $Y$, they are biased by the selection process between $Z$ and $X$, and between $Z$ and $Y$. Therefore, the existence of selection bias is identified.

## E Experimental setting and results on synthetic dataset

In this section, we show more results of GISL compared with baselines and evaluate its robustness. The code of GISL is available at here.

**The accuracy of identifying latent confounders.** Beyond verifying GISL's ability to detect selection bias in Table 3, we evaluate its accuracy in identifying latent confounders on synthetic datasets with n=1500, reported in Table 1. On single-cell gene expression datasets, selection bias is operationalized as the pre–post change in proliferation rate summarized by Z-scores (Section 4.3). By contrast, latent confounders are unobserved and lack a direct interventional response or gold standard, precluding systematic real-world evaluation; accordingly, we assess this component exclusively on synthetic benchmarks.

**SID evaluation metric** The ability of GISL in identifying regulatory relationships is further evaluated by Structural Intervention Distance (SID) as shown in Table 4.

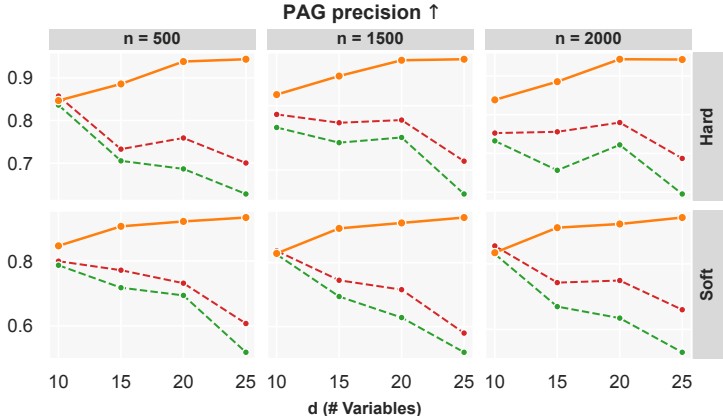

Figure 9: Comparison results in identifying regulatory relations under PAG precision. All values are averaged over 10 runs with different random seeds.

Table 3: Accuracy % of GISL in identifying latent confounders on synthetic data. We report the mean and variance values of accuracy across 10 independent graphs for each configuration with n=1500.

| $|\mathcal{X}|$ | 10 | 15 | 20 | 25 |
|---|---|---|---|---|
| Hard intervention | $62.5 \pm 22.8$ | $63.2 \pm 20.6$ | $68.8 \pm 15.2$ | $66.7 \pm 14.7$ |
| Soft intervention | $66.8 \pm 25.3$ | $68.4 \pm 22.9$ | $70.4 \pm 18.5$ | $71.2 \pm 15.6$ |

Table 4: Experimental results of GISL evaluated by Structural Intervention Distance (SID) on the synthetic dataset under hard intervention

| $|\mathcal{X}|$ | 10 | 15 | 20 | 25 |
|---|---|---|---|---|
| n= 500 | $4.5 \pm 1.5$ | $5.1 \pm 1.2$ | $5.8 \pm 0.9$ | $7.7 \pm 1.0$ |
| n=1500 | $4.3 \pm 1.2$ | $4.8 \pm 1.9$ | $5.6 \pm 1.1$ | $7.3 \pm 1.3$ |
| n=2000 | $4.2 \pm 1.1$ | $4.7 \pm 1.4$ | $5.8 \pm 1.0$ | $7.1 \pm 0.9$ |

Table 5: Robustness evaluation of GISL in noise level (the mean of Gaussian noise $\mu$), the complexity of structures (# edges), the number of variables $|\mathcal{X}|$, and the realism function. Baseline setting: $\mu \sim \mathcal{U}(0,2)$, # edge 15, $|\mathcal{X}| = 15$, hard intervention, and random nonlinear functions.

| Settings | F1 | Precision | Recall | SHD | F1 | Precision | Recall | SHD |
|---|---|---|---|---|---|---|---|---|
| | | n=1500 | | | | n=500 | | |
| Baseline | 69.6±1.9 | 80.8±4.3 | 61.8±1.9 | 5.5±6.5 | 61.6±0.9 | 86.8±0.2 | 50.1±1.7 | 5.8±1.4 |
| $\mu \sim \mathcal{U}(0,4)$ | 72.6±1.8 | 93.5±1.7 | 62.1±3.2 | 4.7±4.6 | 67.4±1.8 | 87.2±2.9 | 56.1±1.7 | 5.7±6.6 |
| $\mu \sim \mathcal{U}(2,4)$ | 77.4±0.8 | 89.7±6 1.4 | 69.2±1.1 | 4.3±3.8 | 67.9±1.0 | 88.8±1.3 | 56.1±1.5 | 5.6±4.0 |
| # edge 20 | 67.6±0.6 | 92.8±0.9 | 59.4±0.9 | 7.2±3.3 | 59.2± 1.3 | 90.1±1.2 | 49.4±1.5 | 8.8±4.7 |
| $|\mathcal{X}| = 25$ | 67.3±0.3 | 85.8±1.2 | 56.1±0.9 | 10.8±1.9 | 64.3±0.6 | 90.8±0.2 | 50.6±1.1 | 11.2±3.6 |
| $|\mathcal{X}| = 50$ | 71.2±1.6 | 90.7±1.3 | 62.4±2.1 | 14.2±6.8 | 68.8±1.4 | 87.2±0.5 | 53.7±0.9 | 16.3±7.0 |
| Normalised-Hill differential equations | 67.6±1.7 | 83.1±2.6 | 59.7±3.2 | 5.9±5.3 | 59.4±1.3 | 85.7±2.3 | 48.9±2.1 | 6.4±3.6 |

**Robustness** Table 5 evaluates the robustness of GISL across noise levels, structural/functional complexity of the Structure Causal Model (SCM), and variable counts. Specifically, increasing the district when sampling the mean of Gaussian noise improves the performance, as bigger distribution changes are easier for CI test tools to detect. Greater structural and functional complexity causes a modest decline in overall performance, whereas holding other factors fixed, sparser graphs yield better performance.

**Overall performance in PAG.** To represent causal relationships in the presence of latent confounders and selection bias, we draw inspiration from ancestral graphs and the notations in [55]. We employ six types of edges, —, →, ↔, ∘—, ∘—∘, ∘→, to represent equivalence classes of conditional

Table 6: Experimental results of GISL and computational baselines on synthetic data

| Methods | Acc | Recall | F1 | SHD |
|---|---|---|---|---|
| GISL | 94.7±0.01 | 95.1±0.01 | 94.9±0.01 | 1.0±0.54 |
| PIDC [11] | 51.6±8.45 | 95.2±0 | 55.1±0.03 | 19.4±12.6 |
| PPCOR [5] | 43.4±10.32 | 97.3±0.28 | 49.7±9.82 | 26.8±7.6 |

independence (CI) patterns. Among these, $\rightarrow$, $\leftrightarrow$, $-$, $\circ\rightarrow$, and $\circ-$ corresponds to the structures illustrated in Figure 3, while undirected edges ($-$) represent there are both latent confounders and selection bias. To assess the ability of GISL in recovering regulatory relationships, selection processes, and latent confounders, Figure 9 compares the accuracy of the PAG learned by GISL against canonical baselines that accommodate selection bias and latent confounding.

All in all, although the specific structures of causal relationships, latent confounders, and selection bias are indeterminate, every pair of nodes shares the same d-separation properties, exhibiting identical conditional independence relations. This leads to Markov equivalence, which can be uniquely represented by a DAG with $\mathcal{L}$ and $\mathcal{S}$. Here, the structures involving $\mathcal{L}$ and $\mathcal{S}$ do not specifically indicate a true common cause or v-structure; rather, they represent the presence of latent confounders or selection bias. However, unlike the output of FCI, the output of GISL is more precise. This is because FCI initializes the causal graph with $\circ - \circ$, which can represent all types of edges. Subsequently, rules are applied based on the limited information provided by v-structures to determine the specific edge types. For instance, while FCI only utilizes v-structures to infer additional edges, as shown in (b) of Figure 7, where $X \circ - \circ Y$ is output, GISL refines this to $X \rightarrow Y$, providing a more accurate representation of the equivalence class of CI patterns. With CI patterns, the equivalence classes are more precise.

**GISL V.S. computational methods under selection bias** We rethink the gene regulatory network inference from a causal view and focus on identifying the causal relationship, latent confounders, and selection process. The setting and output of GISL differ from those of computational methods, which are unable to address the dependencies caused by latent confounders and selection bias. The experimental results of GISL and computational methods on synthetic data are provided for comparison as shown in Table 6. From the table, it is evident that computational methods fail to identify causal relations in the presence of selection bias. This failure occurs because the selection process affects not only the directly targeted variables but also those connected through the same causal pathways. For example, (C) in Figure 7, if perturbing $Y$, the distribution of both $Z$ and $X$ changes. Even without gene perturbation data, computational methods that rely solely on dependence (correlation or co-occurrence) consider these variables as dependent. Consequently, their output often results in a fully connected graph.

**Comparison with CDIS [92]** CDIS models selection as a one-time process that occurs only before intervention and becomes inactive afterward, using an interventional twin graph. Under this assumption, the distribution satisfies $P(Y \mid \text{do}(X) = x, S) = P(Y \mid S)$, meaning that after an intervention on $X$ is applied, the selection variable S no longer influences $Y$. In contrast, our work focuses on biological constraints in genes—a form of inherent selection bias that exists both before and after intervention. These constraints come from basic biological rules, such as essential gene functions or conditions needed for a cell to survive, which do not disappear even when a gene is perturbed. As a result, selection continues to affect the system even after $\text{do}(X)$, violating the CDIS assumption. In our setting, the correct relation is $P(Y \mid \text{do}(X) = x, S) \neq P(Y \mid S)$. This fundamental difference means that CDIS cannot account for such persistent selection effects, leading to poor performance in settings where biological constraints are present, as shown in Table 7.

# F   Experimental settings and results of real-world dataset

**Data availability.** With the advent of next-generation sequencing (NGS) techniques, such as single-cell RNA-sequencing (scRNA-seq), the availability of single-cell data enables a deeper analysis of gene expression in biological systems, offering an unprecedented resolution at the level of individual cells [93]. Moreover, thanks to the advancement and maturation of gene sequencing and perturbation tools, including CRISPR-Cas9 [94], CRISPRi [95], and CRISPRa [96], genes are transformed

Table 7: Comparison with CDIS on synthetic dataset with $|\mathcal{X}|$=15, n=1500, soft intervention. Experimental results are evaluated by F1, Precision, Recall, and SHD in regulatory relationships, and by Precision in identifying selection bias.

| Method | F1 | Precision | Recall | SHD | Precision (selection bias) |
|--------|-----|-----------|--------|-----|----------------------------|
| GISL | $80.0 \pm 1.7$ | $83.9 \pm 3.2$ | $74.1 \pm 1.6$ | $4.1 \pm 5.7$ | $80.8 \pm 11.3$ |
| CDIS | $63.8 \pm 4.5$ | $61.4 \pm 4.7$ | $68.1 \pm 5.3$ | $8.4 \pm 6.4$ | $34.1 \pm 9.1$ |

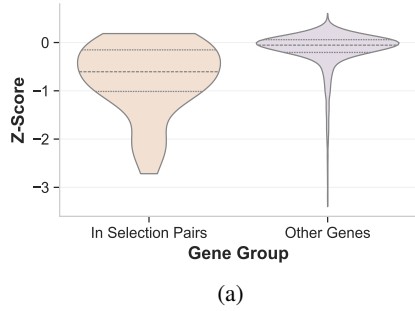 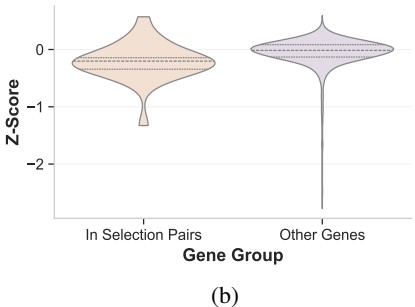

(a)                  (b)

Figure 10: Comparison of Z-score distribution between genes in selection pairs and others on Adamson (a) and Norman (b) datasets.

into viable subjects for causal discovery by providing high-quality single-gene observational and perturbation (interventional) data through the systematic technique Perturb-seq [97, 31, 32].

In real-world datasets, the causal relationships are evaluated based on Enrichr. However, not all perturbed genes are reported in Enrichr, as some genes cannot be perturbed or processed by biological tools like ChIP-Seq. To evaluate the selection process, a z-score is used to verify its existence. The z-score represents the ratio of the growth rate between perturbed genes and normal genes. Changes in growth rates indicate variations in sample size, which align with the characteristics of the selection process. Thus, the z-score serves as an evaluation tool. Distributions of the z-scores for the reported genes are shown in Figure 6. From the figure, we observe that these genes exhibit differences in growth rates between the perturbed and normal cells, indicating the presence of a selection process. In some cell lines, the growth rate does not change, suggesting that the gene is not under selection in those cells. This observation is consistent with our explanation regarding differential gene expression.

### F.1 Experimental results on real-world datasets

We conduct experiments on three representative datasets of single-cell gene expression in the real world to verify the effectiveness of our proposed method, including data from K562 cells [32, 61] and Human Lung Epithelial Cells (HLEC) [31]. Firstly, the skeleton is discovered among 5012 genes. Then, the structure of the perturbed genes is identified. Since GISL outputs an equivalence class, the structures represent CI patterns. For example, $\rightarrow$ denotes the equivalence class corresponding to the first CI pattern illustrated in Figure 3. The edge $-$ indicates the indeterminate edges usually under selection, characterized by CI patterns where $I$ remains conditionally dependent on observed genes, regardless of the conditioning set. The selection variable $S$ and latent confounder $L$ specifically indicate the presence of selection bias or latent confounders, respectively.

The experimental result on the Dixit dataset is shown in Figure 5. Where blue edges are verified as correct by Enrichr, where the ChEA 2022 [98], TF-gene enrichment [99], and [100] are mainly considered. We consider them to be more reliable because ChIP-Seq, used in ChEA, directly detects the binding sites between transcription factors and genes. Additionally, TF-gene enrichment identifies robust pairs supported by enrichment analysis. Moreover, the selection bias is supported by the high z-scores observed for the genes such as $RACGAP1$ (-0.903), $E2F4$ (-0.158), $GABPA$ (-0.543), and $NR2C2$ (-0.127), respectively. Experimental results indicate that almost all directed causal

relationships discovered by GISL are correct, while undirected edges remain undetermined. This is because undirected edges suggest the presence of both selection bias and latent confounders, making the causal relationship unidentifiable. This is because, with selection bias and latent confounders, the CI pattern becomes fully dependent, which obscures the unique characteristics of causal relationships in the CI pattern. The experimental results of PPOCR are shown in Figure 12. The results show that our proposed GISL demonstrates greater accuracy and reliability in matching the results of biological experiments. In addition, the experimental results of GISL on the Norman and Adamson datasets are shown in Figure 11 and Figure 13. Correspondingly, the Z-score distribution comparison between genes under selection bias and others is illustrated in Figure 10.

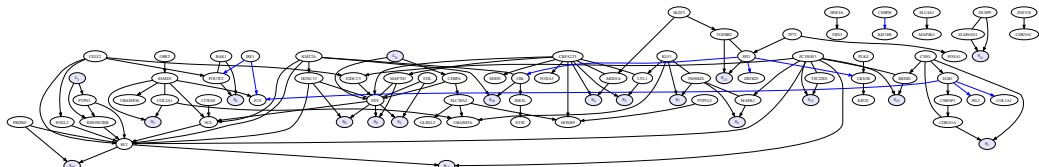

Figure 11: Experimental result of GISL on the Norman dataset.

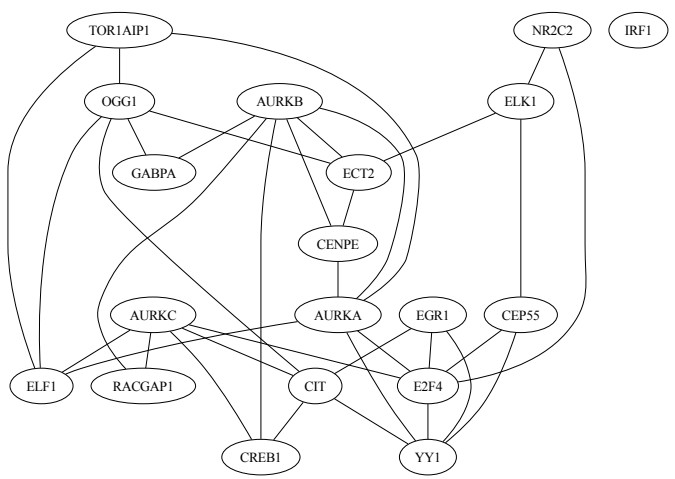

Figure 12: Experimental result of PPCOR on the Dixit dataset.

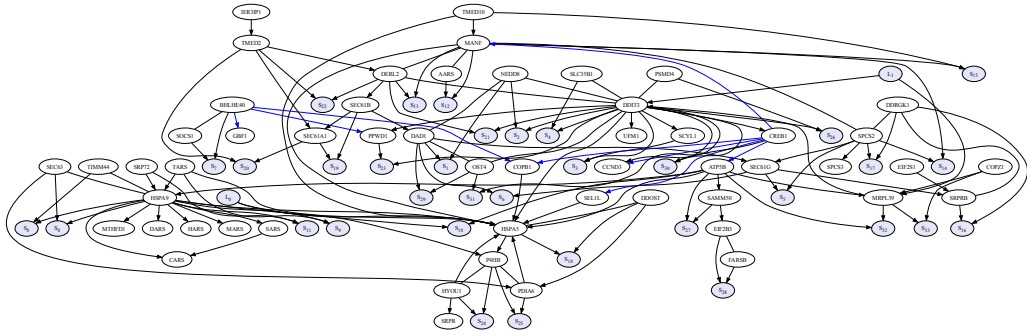

Figure 13: Experimental result of GISL on the Adamson dataset.

Moreover, biological analysis for the human lung epithelial and k562 lukemia cell lines is discussed as follows:

**Human Lung Epithelial Cells**    GISL reports that there exist biological constraints between gene CNN1 and CDKN1A. CNN1 (Calponin 1) is normally low in healthy lung epithelium but markedly increases during epithelial–mesenchymal transition or fibrotic remodeling [101]. CDKN1A (p21) is

a cell-cycle inhibitor that surges under DNA damage or stress (e.g., p53 activation), causing growth arrest or senescence; indeed, lung injury models and fibrotic lungs (IPF) show abnormally high p21 in alveolar epithelial cells [102]. These observations imply that when lung epithelial cells are pushed toward a damaged or transitioning state, both CNN1 and p21 tend to rise, which suggests that the cell may need to keep its combined activity in check to maintain normal function. Excessive co-elevation of CNN1 (indicating mesenchymal/fibrotic shift) and CDKN1A (indicating cell-cycle arrest) could drive cells into an unsustainable state (senescence or fibrosis), so a balance in their expression is likely required to preserve viability and tissue homeostasis. This suggests a form of biological constraint whereby lung epithelial cells limit the concurrent upregulation of CNN1 and CDKN1A to avoid tipping into pathological remodeling or loss of proliferative capacity.

**K562 Lukemia Cells**   GISL reports that there exist biological constraints between gene ELF1 and gene GABPA. ELF1, an ETS-family transcription factor, regulates hematopoietic differentiation and immune genes; in K562 cells, it controls the G1 to S phase transition via CDKN1A (p21), with overexpression inducing apoptosis and underexpression promoting unchecked proliferation [103]. GABPA, a nuclear respiratory factor, orchestrates mitochondrial biogenesis and metabolism through the PI3K/Akt/mTOR axis; its depletion triggers G0 to G1 phase arrest, and its binding affinity (pKD) and phosphorylation response ($pEC_{50}$) underscore a requirement for dosage matching downstream signaling intensity [104]. Both factors converge on stress and apoptotic pathways, GABPA targets (Caspase-9, Bcl-2) overlap with ELF1, regulated DNA-damage genes, and respond coordinately to PI3K/Akt perturbations (e.g., LY294002) and autophagy induction (e.g., in K562 and ADM cells), suggesting K562 cells impose a constraint on ELF1 and GABPA expression to maintain proliferative homeostasis and survival [105, 106].

# G   Understanding Z-score

Z-score is a result of a model proposed by [30], which is designed to describe the dynamic proliferation process. For single-guided RNAs (SgRNA) $j$ targeting gene $g$ in cell line $c$, the number of cells $N_{cj}(t)$ with sgRNA at time $t$ after perturbation is modeled as follows:

$$N_{cj}(t) = N_{cj}(0) \left( \boldsymbol{p_{cj}} e^{R_{cg}^* t} + (1 - \boldsymbol{p_{cj}}) e^{R_c t} \right), \tag{2}$$

where $t = 0$ is the time of perturbation, $p_{cj}$ is the probability that the sgRNA $j$ achieves knockout of its target in cell line $c$, $R_c$ is the unperturbed growth rate of the cell line, and $R_{cg}^*$ is the new growth rate caused by knockout of the targeted gene in the given cell line. The gene fitness effect is the fractional change in growth rate $r_{cg} = R_{cg}^*/R_c - 1$. Gene fitness effects are the primary desired output for this type of experiment.

Considering the delayed perturbation effect, the model built in practice to estimate the number of cells are as follows:

$$v_{cj}^L(t > d_g^L) = v_{cj}^L(0) \left( 1 + p_c^L p_j \left( e^{R_c^L r_{cg}(t - d_g)} - 1 \right) \right) / Z_c^L(t) \tag{3}$$

where

- $c$ indexes cell line, $j$ indexes sgRNA, $g$ indexes gene, $L$ indexes library or batch, and $t$ is the time elapsed since library transduction
- $v_{cj}^L(t)$ is the model estimate the normalized readcounts of sgRNA $j$ in cell line $c$ screened in batch or library $L$ at time $t$
- $v_{cj}^L(0)$ is the model estimate of the normalized number of cells initially receiving sgRNA $j$
- $p_c^L$ and $p_j$ are the estimated CRISPR knockout efficacies in cell line $c$ with sgRNA $j$
- $R_c^L$ is the estimated unperturbed growth rate of the cell line
- $r_{cg}$ is the estimated relative change in growth rate for that cell line if gene $g$ targeted by sgRNA $j$ is completely knocked out
- $d_g$ is the delay between infection and the onset of the growth phenotype
- $Z_c^L(t)$ is a normalization equal to the sum of the numerator over all sgRNAs $j$ in the cell line for the given library and time point

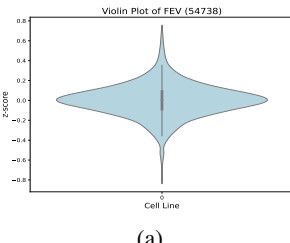
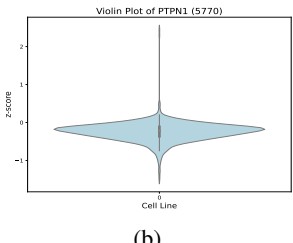
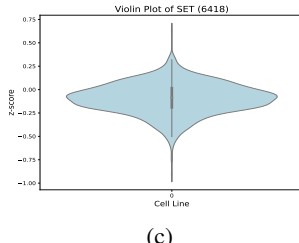

| (a) | (b) | (c) |

Figure 14: The distribution of z-scores for the genes FEV, PTPN1, and SET across all cell lines highlights the variation in sample size.

The Z-score describing the ratio of growth rate before and after perturbation is designed as follows:

$$Z_{cj}(t) = v_{cj}(0) \left( 1 + p_c p_j \left( e^{R_c r_{cg}(t-d_g)} - 1 \right) \right) \quad \forall t \geq d_g \tag{4}$$

$$Z_{cj}(t) = v_{cj}(0) \quad \forall t < d_g. \tag{5}$$

Using FEC, PTPN1, and SET as representative genes, the calculated Z-scores across a panel of cell lines are shown in Figure 14. Cell lines with a Z-score of zero indicate the absence of selection constraints on the genes: perturbation does not induce measurable cell death, and the expression distribution remains unchanged. In contrast, non-zero Z-scores mark genes under selection constraints, where perturbation produces pronounced shifts in their expression profiles. Moreover, each gene displays a distinct Z-score pattern across the cell-line panel, reflecting gene-specific selective dynamics. These results align with the differential gene-expression patterns expected under selective pressure.

## H Discussion

**Discussion and Limitations.** In cells, we argue that the different intracellular environments, acting as selection mechanisms, restrict the expression of genes. When the environment remains, a selection mechanism is always present. Genes stay in cells with the remaining environment, showing the reasonability of our setting. The superiority of GISL is that it can identify the existence of selection bias and latent confounders, as well as causal relationships. Despite its flexibility in integrating observational and perturbation data within a unified causal framework, a fundamental limitation remains: whenever selection bias is present, spurious marginal and conditional dependencies inevitably arise (see Section 3.4). Consequently, reliably detecting selection in the presence of such bias remains an outstanding challenge.

**More Discussion.**
**Q1: Why does the paper explicitly look at gene regulatory networks when the method is more generally for causal discovery with interventional data?**
**A1:** GISL is broadly applicable to general causal discovery, but we chose the gene regulatory network inference (GRNI) framework because it was inspired by a real-world challenge, the pervasive yet often overlooked selection (survival) bias in biological data, and directly addressing this issue offers immediate value for GRNI and beyond.

Specifically, we focus on GRNI because:

- GRN inference is a classical causal discovery task with strong biological significance, supported by well-established gene perturbation technologies that provide abundant interventional data.
- Despite being a classical problem, modeling gene expression from a causal perspective remains challenging due to latent confounding and overlooked selection bias, making it a meaningful testbed for our method.

**Q2: Why GISL performs the best regarding the recall? A2:** As shown in Figure 4, GISL shows a little bit lower recall in identifying regulatory relationships under selection bias. This is because, in theory, it is challenging to distinguish (c): selection bias and (e): causal + selection bias, as they share the same CI patterns (all conditionally dependent). As a result, our GISL categorizes both cases

(c) and (e) as selection bias and thus shows lower recall when identifying regulatory relationships (causal relations).

On the other hand, the baseline algorithm, e.g., GIES, exhibits higher recall, but this does not necessarily indicate better performance. The key issue is that GIES is unable to account for selection bias, which introduces spurious dependencies into the data. When GIES relies on a score function to detect conditional dependencies, it may mistakenly interpret these spurious associations as causal relationships. As a result, the algorithm tends to include additional edges regardless of whether true causal relationships exist, leading to inflated recall but low accuracy.

