# OpenReview forum: "Gene Regulatory Network Inference in the Presence of Selection Bias and Latent Confounders"
_NeurIPS.cc/2025/Conference — NeurIPS 2025 poster_

### Official Review · Reviewer_7xy2 · 2025-06-28

**Clarity:** 1
**Significance:** 2
**Originality:** 4
**Rating:** 3
**Confidence:** 1

**Summary:**

This paper focuses on gene regulatory network inference (GRNI), where the goal is to identify causal relationships between gene expression (how one gene promotes or represses the expression of another gene in a cell) based on measurements of gene expression levels observed in cells. The authors highlight that the set of cells observed can represent a significant selection bias; two genes may appear to have high levels of co-expression not because of any specific regulatory relationship but rather because cells with both genes are more likely to be selected for sequencing (“satisfying the survival criteria”).

The authors point out a lack of methods for GRNI that address this selection bias. They then note that unlike bias arising from hidden confounders, selection bias is symmetric: meaning that perturbing gene 1 influences a behavior in gene 2, and perturbing gene 2 also influences gene 1. They propose a novel method, gene regulatory network inference n the presence of selection bias and latent confounders (GISL) that exploits this symmetry to tolerate selection bias when performing GRNI.

Contributions:
- The authors propose a novel method for GRNI in the presence of selection bias, and claim consistency results under minimal assumptions.
- The authors draw attention to the issue of selection bias and highlight some tractable properties (symmetry) of selection bias.
- The authors conduct experiments on both synthetic and real-world data to compare their novel procedure to baselines.

**Questions:**

Regarding Figure 3: What do the set of symbols (up/down triangles, diamonds, circles in black and white) refer to, and what does it mean for them to be “green check” versus “red X” in the reference table? Are these symbols meant to show that certain conditional independence relationships are/aren’t satisfied in each of the columns? Is it important that columns c and e grey-shaded, and is that meant to communicate something?

As explained in Weaknesses, I did not understand the discussion of mutilated DAGs and I did not understand how they related to the rest of the paper. Could the authors briefly explain?

In Figure 4, GISL clearly out-performs the baselines in F1, precision, and SHD, but achieves only middling results in recall. Do the authors have any hypotheses for why GISL would perform so well with regards to the other metrics, but not with respect to recall?

**Ethical Concerns:**

["NO or VERY MINOR ethics concerns only"]

**Final Justification:**

I maintain my score (borderline reject) because I do not believe the core contributions of this paper are explained in sufficient clarity in the current state of the manuscript. I explained my reasoning to the authors in my comments, and I reiterate my main concern here: multiple reviewers highlighted confusion about Figure 3. Because the main novel method proposed in this work (Algorithm 1) is defined only through reference to Figure 3, in my opinion, it is necessary that Figure 3 be understandable to readers and completely correct. Personally, I am not confident in my level of understanding of the algorithm and thus cannot be confident in my ability to verify the consistency results. I mark my confidence level at the lowest available (1) to reflect this.

**Limitations:**

yes

**Paper Formatting Concerns:**

I have no issue with the formatting. However I do take issue with the low level of explanation dedicated to Figure 3, given its use to define the core novel algorithm.

**Quality:**

2

**Strengths And Weaknesses:**

Strengths:
- The authors make a compelling case that selection bias is a serious obstacle to causal discovery and that it is under-addressed.
- The paper contains several examples that illustrate concepts at a high-level. Section 2.2 and Figure 2 do an excellent job of communicating spurious dependencies arising from selection bias.
- The authors claim consistency results for their procedure under minimal (and very typical) assumptions.

Weaknesses:

In several places, the technical content is extremely dense and terms/quantities appear without definition. This seriously limited my ability to understand central parts of the paper, such as the pseudocode of Algorithm 1, the novel proposed method.
- Figure 3 looks very detailed but I could not interpret it at all. Unfortunately this is an essential figure to understand, because the novel method proposed (Algorithm 1) is defined by referencing sub-panels of Figure 3. I list some clarifying questions about Figure 3 in the Questions section of the review.
- The preliminaries are full of undefined technical terminology, e.g. mutilated DAG. If these are standard terms within GRNI and the other reviewers disagree with my assessment then perhaps this is not a significant weakness, but as a reader I found this very confusing. These terms are not defined in the main body nor in Appendix A, which is designated for definitions of core concepts. Some terms are explained indirectly much later in the paper (e.g. knockout vs knockup and knockdown for hard/soft interventions), but mutilated DAGs are never referenced again and I still do not understand what they represent nor why they were presented in the preliminaries.
- I found several sentences un-parsable. E.g. Line 102, “For hard interventions, consider each $T_k$ as factoring in a mutilated DAG…”. $T_k$ is not defined, and given the general lack of definitions I found it impossible to decipher what $T_k$ referred to based on context. Is it an element of $\cal{T}$? I thought $\cal{T}\subseteq \cal{X}$ so I expected elements of $\cal{T}$ to be denoted $X_k$.


I caveat this by stating that I am not familiar with definitions relating to interventions, so if other more experienced reviewers find the paper reasonably readable then I defer to their opinions.

---

> ### Author Rebuttal · Authors · 2025-07-30
>
> Firstly, we would like to sincerely thank you for the considerable time and effort invested in reviewing our paper. Your insightful comments and suggestions have significantly helped us enhance the clarity and readability of our manuscript. In response to your valuable feedback, we have carefully revised and clarified our paper:
>
> ---
>
> **Q1: What do the symbols and reference table  mean in Figure 3?**
>
> We add a detailed explanation for Figure 3 to clarify the notation.
>
> The set of symbols (up/down triangles, diamonds, and circles) represents the four types of conditional independence (CI) relationships listed in the first column of the reference table in Figure 3(f). Each symbol corresponds to a specific CI pattern. The  **green check** indicates that the CI relationship is **satisfied** and is marked with corresponding **black symbols**,  while the **red X** indicates that the CI relationship is **not satisfied** and is marked with **white symbols**. For example, a black upward triangle indicates that $I_X \perp Y \mid S$ holds (i.e., I_X and Y are conditionally independent given S), whereas a white upward triangle indicates that $I_X \not\perp Y \mid S$ (i.e., they are dependent). We hope this clarification makes the figure more accessible and easier to interpret.
>
> **Taking Figure 3(a) as an example,**  the structure exhibits conditional dependence $Y \not\perp I_X \mid S$, which is represented by the white upward triangles (△) in the reference table. The conditional independence $Y \perp I_X \mid X, S$ is shown in (a) when $X$ is given and is represented by black downward triangles (▼). These CI patterns for (a) are summarized using the corresponding symbols displayed below the causal graph.
>
> **Q2: What do the grey-shaded columns (c) and (e) mean in Figure 3?**
>
> The grey shading in columns (c) and (e) indicates that both cases yield identical conditional independence (CI) patterns. Specifically, as discussed in Remark 3.4, the selection process introduces additional conditional dependencies into the observed data, resulting in indistinguishable CI patterns for these two scenarios. Consequently, GISL is unable to differentiate a scenario involving only selection bias (c) from one involving both selection bias and a causal relationship (e). However, this limitation does not impact GISL's capability to identify the presence of selection bias itself, even though the precise causal structure remains ambiguous. This point is formalized in Theorem 3.7. The grey shading thus serves to distinguish scenarios that GISL can uniquely identify (white shading: cases a, b, and d) from those it cannot (grey shading: cases c and e, which are indistinguishable).
>
> Please let us know if you have any questions about Figure 3. We would be delighted to discuss this further and deeply appreciate your thoughtful feedback.
>
> ---
>
> **Q3: Technical terminologies & terms are Undefined: mutilated DAG, knockout, knockup, and knockdown.**
>
> We sincerely appreciate your careful reading and thoughtful comments. We are grateful for the opportunity to clarify these terms and improve the clarity of our presentation.
>
> To enhance readability, we have added explicit definitions for commonly used terms introduced in the preliminaries, including mutilated DAG (widely used in interventional causal discovery), perturbation indicator (Definition 1 on page 6 of [1]), intervention targets (third paragraph on page 4 of [2]), and hard/soft interventions (illustrated in the figures on pages 4 and 6 of [3]).
>
>
> **Why introduce Mutilated DAG?**
> _Mutilated DAG_ represents the graphical structure for hard intervention (illustrated in Figure 2(a) and described in Lines 102–104), while _Augmented DAG_ is introduced in Lines 105–108 to represent soft interventions. Since our framework accommodates both types of interventions, it is necessary to present both forms of DAG representations.
>
>
> **Logical progression in introducing terms.**
> We have aimed to reduce cognitive load for the reader by introducing terminology in a gradual, structured manner. In our preliminaries (Lines 100–102), we **first** introduce gene perturbation methodologies—**gene knockout and transcriptional modulation** (gene knockup or knockdown)—which correspond to hard and soft interventions, respectively, in the causal inference framework. We then describe how these two types of interventions are represented graphically. Specifically, a **mutilated DAG** refers to the graph obtained from a causal DAG by **removing all incoming edges to the intervened variables**, effectively breaking the causal dependence of those nodes on their parents. Similarly, the **augmented DAG** is introduced in Lines 105–108 to represent soft interventions, and is illustrated in Figure 2(b).
>
> > [1] Varieties of causal intervention. PRICAI 2004.
>
> > [2] Characterization and greedy learning of interventional Markov equivalence classes of directed acyclic graphs. JMLR 2012.
>
> > [3] Interventions and causal inference. PSA 2006.
>
> ---
>
> **Q4: $T_k$ in the sentences is Unparsable.**
>
> Each $T_k$ denotes the target variable set for the k-th intervention. We have added the definition of $\mathcal{T} = \{T_1, \ldots, T_K\}$ in the revised version.
>
>
> ---
>
> **Q5: Why GISL performs not so well regarding the recall?**
>
> As you pointed out, GISL shows a little bit lower recall in **identifying regulatory relationships** under selection bias. This is because, as discussed in **Q2** and Figure 3, in theory, it is challenging to distinguish `(c): selection bias` and `(e): causal + selection bias` -- they share the same CI patterns (all conditionally dependent). As a result, our GISL categorizes both cases `(c)` and `(e)` as selection bias and thus shows lower recall when identifying regulatory relationships (causal relations).
>
> On the other hand, the baseline algorithm, e.g., GIES, exhibits higher recall, but this does not necessarily indicate better performance. The key issue is that GIES is unable to account for selection bias, which introduces spurious dependencies into the data. When GIES relies on a score function to detect conditional dependencies, it may mistakenly interpret these spurious associations as causal relationships. As a result, the algorithm tends to include additional edges regardless of whether true causal relationships exist, leading to inflated recall but low accuracy.
>
>
> ---
> We hope our responses have clarified the key points and improved the overall clarity of the paper. If any aspects remain unclear, we would be happy to provide further explanation. Thank you again for your thoughtful review and the considerable effort you invested.

---

> ### Comment · Reviewer_7xy2 · 2025-08-03
>
> I thank the authors for their responses. I am still confused about how to parse the main algorithm (Algorithm 1) using Figure 3. For example in step 3, can the authors explain in prose exactly when the condition "If $\mathcal{J}== \operatorname{Figure 3a}$" is triggered?
>
> Edit: correcting LaTeX formatting.

---

> ### Author Response · Authors · 2025-08-04
>
> Thank you for the follow-up.  **Our core idea is that different causal graphs lead to different patterns of which variables are independent or dependent.** We can use conditional independence (CI) tests to check these patterns. By testing the CI relations between gene-perturbation indicators ($I_X$ and $I_Y$ in Figure 3) and the observed variables ($X$ and $Y$ in Figure 3), some fundamental causal structures in the presence of selection bias and latent confounders can be uniquely identified.
>
> **In prose:**
>
> In Step 3 of Algorithm 1, for any two variables ($X_i, X_j$) (corresponding to ($X, Y$) in Figure 3), we apply conditional independence (CI) tests to see how each perturbation variable ($I_i, I_j$) relates to the other observational variable given appropriate conditioning.  For example, we can observe the following CI relations hold from the data $\mathcal{J} =$ \{$I_{i} \not\perp X_{j} \mid \mathcal{S}$, $I_{i} \perp X_j\mid X_i, \mathcal{S}$, $I_{j} \perp  X_i \mid \mathcal{S}$, $I_{j} \not\perp X_i \mid X_j, \mathcal{S}$\}. We then compare these four independence/dependence results to the CI relations revealed by the causal graph of Figure 3(a).  If and only if those four outcomes line up perfectly with what Figure 3(a) prescribes, we trigger the “If $\mathcal{J}==Figure 3(a)$” condition and conclude $X_i \to X_j$.
>
> ---
>
> **Kindly note that the OpenReview system may occasionally fail to render LaTeX correctly. If you encounter any unrendered LaTeX, please refresh the page. Thank you.**

---

> > ### Author Response · Authors · 2025-08-06
> > **Could you please kindly let us know if our further response answers your questions?**
> >
> > Dear Reviewer 7xy2,
> >
> > We are very grateful for your careful review and engaged discussion. In response to your question, we have provided a prose explanation of how Algorithm 1 maps to the CI patterns illustrated in Figure 3, using Figure 3(a) as a concrete example to clarify when the condition “$\mathcal{J} =$Figure 3(a)'' is triggered (see our updated response above). We hope this explanation resolves the remaining confusion.
> >
> > If there are any further concerns or if additional clarification would be helpful, we would be happy to elaborate further. We sincerely appreciate your time and thoughtful feedback throughout the review process.
> >
> > Yours sincerely,
> >
> > Authors of Submission 17977

---

> > > ### Comment · Reviewer_7xy2 · 2025-08-06
> > >
> > > I thank the authors for their engaged responses. I now better understand Figure 3.
> > >
> > > I will maintain my score: I explain my core reasoning below.
> > >
> > > Multiple reviewers highlighted confusion about Figure 3. Because the main novel method proposed in this work is defined only through reference to Figure 3, in my opinion, it is necessary that Figure 3 be understandable to readers and completely correct. Given the difficulty of parsing this figure, the error in Figure 3 (which implies an error in the main algorithm) caught by Reviewer TNYS, and the error in the way the columns of Figure 3 are referenced caught by Reviewer 7UVK, I do not have high confidence in recommending this work to readers at this stage, and so I maintain my score.

---

> > > > ### Author Response · Authors · 2025-08-06
> > > >
> > > > Thank you for your engaged follow-up. We are happy that you can better understand Figure 3 with our response.  Below, we provide the point-to-point response to the remaining concerns.
> > > >
> > > > > The error in Figure 3 (which implies an error in the main algorithm), caught by Reviewer TNYS,
> > > >
> > > > Thank you for highlighting this and helping us improve the figure’s clarity. We respectfully note that, as detailed in our response to Reviewer TNYS (Q7), we have corrected the inverted symbol color in the top panel of Figure 3(b), and Reviewer TNYS has acknowledged that this issue is addressed. Specifically,
> > > >
> > > >    The intended CI relations are $I_X \perp Y \mid \mathcal S$ and $I_X \not\perp Y \mid X,\mathcal S$, which should correspond to the black upward triangle (▲) and the white downward triangle (▽), respectively. We corrected the flipped symbols by reversing their colors.
> > > >
> > > > > the error in the way the columns of Figure 3 are referenced caught by Reviewer 7UVK,
> > > >
> > > > Regarding the flipped references to Figures 3(c) and 3 (d), we have corrected the swapped references.
> > > >
> > > > if $\mathcal{J} == (c)$ then $S \leftarrow (X_i, X_j)$.
> > > >
> > > >  if $\mathcal{J} == (d)$ then $L^{'} \leftarrow (X_i, X_j)$
> > > >
> > > >
> > > > We totally agree that the figure should be both correct and clear to ensure readability. To achieve this goal, in the revised version, we have clarified the relevant details and corrected the flipped symbols and references. To further improve the figure, we would be grateful if you could elaborate on which aspects remain difficult to parse.
> > > >
> > > > Thank you again for your valuable suggestions on Figure 3, which have helped us clarify details and improve the paper’s readability.

---

> ### Comment · Reviewer_7xy2 · 2025-08-08
>
> Re: "To further improve the figure, we would be grateful if you could elaborate on which aspects remain difficult to parse." In general, I think the current way the figure is used to define the method leaves too many steps for the reader to reconstruct on their own. For example, in their algorithm, the authors define $\mathcal{I}$ as a set of 4 independence tests, and then reference logical conditions like "If $\mathcal{I}==\operatorname{Figure}\ 3(a)$," where Figure 3(a) displays two in/dependence relationships. As the authors explain in their reply above, what they intend is for the reader to use symmetry to deduce that Figure 3(a) IMPLICITLY references 4 in/dependence relationships obtained by exchanging $X_i$ and $X_j$. This is the kind of detail that is obvious to the authors, but unclear to readers, especially given the fact that even understanding that Figure 3(a) encodes 2 in/dependence relationships is difficult because it is communicated with black/white triangles, diamonds, and circles rather than explained anywhere in prose. The novel method in Algorithm 1 is a key contribution of the paper. Given that readers struggle to understand Algorithm 1's statement without the ability to ask specific questions of the authors (as evidenced by multiple reviewers' confusion), I think the paper offers limited benefits to the community in its current form.
>
> I suggest the authors either restructure Figure 3 so that it is correct and readable or provide precise pseudocode for defining Algorithm 1.
>
> Edit: formatting quotations in markdown.

---

> > ### Author Response · Authors · 2025-08-08
> >
> > Thank you very much for sharing your remaining concerns with us and your engaged follow-up.
> >
> > > "In their algorithm, the authors define I as a set of 4 independence tests, and then reference logical conditions like "If $I==Figure 3(a)$," where Figure 3(a) displays two in/dependence relationships."
> >
> > Let us respectfully add that Figure 3(a) displays four in/dependence relationships (two at the top and another two at the bottom ) rather than two. Specifically, for example,
> > **Figure 3(a) top** shows two CI relationships, △ and ▼, in the box below it:
> >
> > - The conditional dependence $Y \not\perp I_X \mid S$, which is represented by the white upward triangle (△) in the reference table given in Figure 3(f).
> >   - The conditional independence $Y \perp I_X \mid X, S$, which is represented by the black downward triangle (▼).
> >
> > **Figure 3(a) bottom** shows another two CI relationships, $\blacklozenge$ and ○:
> >
> > - The conditional independence $X \perp I_Y \mid S$, which is represented by the black diamond ($\blacklozenge$) in the reference table.
> > - The conditional dependence $X \perp I_Y \mid Y, S$, which is represented by the white circle (○).
> >
> > ---
> > > "As the authors explain in their reply above, what they intend is for the reader to use symmetry to deduce that Figure 3(a) IMPLICITLY references 4 in/dependence relationships obtained by exchanging $X_i$ and $X_j$."
> >
> > Thanks for your thoughtful comment. We wish to clarify that we did **NOT** intend to require readers to infer additional CI relations "by symmetry".  Instead, in Figure 3, we **explicitly** present all four CI relationships matched with Step 3 of Algorithm 1 (please also refer to the response above). There is no additional deduction requirement of swapping $X_i$ and $X_j$.
> >
> > Regarding "symmetry", we only refer to perturbation symmetry, the property of the graphical constraint that we utilized to design the algorithm.
> >
> > ---
> > > "Understanding that Figure 3(a) encodes 2 in/dependence relationships is difficult because it is communicated with black/white triangles, diamonds, and circles rather than explained anywhere in prose."
> >
> > Thanks for your comment. We completely agree that relying solely on symbols (upward/downward triangles, diamonds, circles) made Figure 3 difficult to parse without accompanying prose. In response to the reviewers’ valuable feedback (Q7 of reviewer TNYS, Q1 of reviewer 7xy2), we have addressed the concerns by adding a more detailed description of the reference table (f) of Figure 3. **We have updated the caption of Figure 3 with an illustrative example to improve clarity and readability as follows:**
> >
> > **Figure 3:** Distinguishing causal, selection, and confounding structures via perturbation effect and symmetry. Differences in symmetry and structure: Causation vs. Selection vs Confounding. $\textcolor{#EAE0F1}{\rule[-0.2ex]{2em}{1.6ex}}$  indicates the targeted gene pairs of CI test. (a) refers to the direct causal structure between X and Y (represented by \`C'). (b) means there is a latent variable between them (\`L'). (c) is the selection structure (\`S'). (d) stands for causal and latent relations at the same time (\`C \& L'). (e) stands for causal and selection relations at the same time ('C \& S'). $\textcolor{#F7E1CE}{\rule[-0.2ex]{2em}{1.6ex}}$  contains CI results. `(f) serves as a reference table summarizing the CI patterns for each target gene pair: different symbols correspond to different CI relations; black symbols (▲, ▼, ◆, ● )  indicate the conditional independence, while white symbols (△  ▽  ◇ ○) indicate the conditional dependence. For example, (a) encodes four CI relations: $Y \not\perp I_X \mid S$ (△)  and $Y \perp I_X \mid X, S$ (▼) at the top; $X \perp I_Y \mid S$ (◆) and $X \perp I_Y \mid Y, S$ (○) at the bottom.
> >
> > ---
> >
> > In summary, in light   of your suggestions, we have updated the revision as follows:
> >
> > 1. Reconstruct Figure 3(b) top by correcting the flipped symbol color.
> > 2. Expand the caption of Figure 3 with an illustrative example to more clearly describe the reference table.
> > 3. Update the pseudocode by correcting the swapped references in Algorithm 1.
> > if $\mathcal{J} == (c)$ then $S \leftarrow (X_i, X_j)$.
> > if $\mathcal{J} == (d)$ then $L^{'} \leftarrow (X_i, X_j)$.
> >
> > Thank you again for your valuable suggestions and feedback on Figure 3. We hope you find the revised version clear enough. If you have further thoughts, please kindly let us know.

---

> > > ### Comment · Reviewer_7xy2 · 2025-08-08
> > >
> > > I thank the authors for their further clarification. I will maintain my score, and I retain my confidence score (1) to reflect my own difficulty in parsing Algorithm 1 (stemming from difficulty in interpreting Figure 3).

---

> ### Author Response · Authors · 2025-08-08
>
> Dear Reviewer 7xy2,
>
> We appreciate your time and effort. Your patience and feedback on Figure 3 were invaluable in guiding revisions that substantially improve the paper’s clarity. We have updated Figure 3 with an extended caption and an illustrative example to enhance readability, and we hope these changes address your concerns. Thank you again for your engagement and constructive input.
>
> Sincerely,
>
> Authors of #17977

---

### Official Review · Reviewer_7UVK · 2025-06-29

**Clarity:** 2
**Significance:** 2
**Originality:** 2
**Rating:** 4
**Confidence:** 2

**Summary:**

This paper proposes a gene regulatory network inference algorithm to learn the causal regulatory relationship from gene expression data. The gene expression data involves both hard and soft interventions. The paper examines a generalized intervention setting that incorporates the presence of a latent confounder and selection bias. The main idea is to use the asymmetry of intervention with selection bias to indicate the presence of a regulatory relationship.

**Questions:**

- I don't fully understand the symbols in Figure 3. Could you please explain that in more detail with the reference table?
- This is related to the previous question. What are the terms like $[I_i,X_j|S]$? Are those indicator variables for the conditional independence?
- How scalable is the proposed method? Since the proposed method involves conditional independence testing, how sensitive is the proposed method to the size of datasets?


Some minor details:
- The option in Algorithm 1 $\mathcal{J}==(c)$ and $\mathcal{J}==(d)$ seems to be flipped and inconsistent with the caption of Figure 3.
- Do the terms regulatory relations and causal relations mean the same thing in this context? The paper seems to use these terms interchangeably.

**Ethical Concerns:**

["NO or VERY MINOR ethics concerns only"]

**Final Justification:**

In the rebuttal, the authors address my primary concern about the clarity of the paper and comparison with existing works. I have raised my score accordingly. The reason for not giving a higher score is that I think the paper could still be improved with better presentation of the problem setup and the methodology.

**Limitations:**

yes

**Paper Formatting Concerns:**

No paper formatting concerns.

**Quality:**

3

**Strengths And Weaknesses:**

# Strengths

- The paper studies an interesting and relevant problem.
- The proposed method is evaluated with both synthetic and real-world data experiments.

# Weaknesses

- Some of the notation and explanation in the paper seems a bit confusing (please see questions below).
- The paper does not compare with some existing work with a very similar setting (such as Dai et al. 2025).
- The problem setting of the selection variable is somewhat vague. The symmetry of the selection process requires the selection variable to take different values under intervention. Thus, in a sense, the selection bias is mitigated by introducing the intervention variable in the augmented graph. This is a bit different from the standard interpretation of selection bias in the observational setting. Maybe this deserves a more careful discussion.


Dai, Haoyue, et al. "When Selection Meets Intervention: Additional Complexities in Causal Discovery." The Thirteenth International Conference on Learning Representations.

---

> ### Author Rebuttal · Authors · 2025-07-31
>
> Thank you for the time dedicated to reviewing our paper. Your insightful comments and valuable feedback have greatly improved its clarity. Following your suggestions, we have carefully revised the manuscript. Please see our point-by-point responses below.
>
> ---
> **Q1:** **"I don't fully understand the symbols in Figure 3. Could you please explain that in more detail with the reference table?"**
>
> We have added a detailed explanation of Figure 3 with the reference table to improve clarity.
>
> We begin by explaining the reference table in Figure 3(f). The **first column** lists key conditional independence (CI) patterns used to distinguish different causal structures. **Each pattern is represented by a specific symbol—such as upward/downward triangles, diamonds, and circles—corresponding to different types of CI relations (e.g., between $I_X$ and $Y$).**
>
> A **green check** indicates that a CI relation **holds**, shown by a **black symbol**. Conversely, a **red X** indicates that the CI relation **does not hold** (i.e., conditional dependence), represented by a **white symbol**. This symbolic representation helps visually summarize the CI patterns associated with each structure in Figure 3.
>
> **Taking Figure 3(a) as an example,**  the structure exhibits conditional dependence $Y \not\perp I_X \mid S$, which is represented by the white upward triangles (△) in the reference table. The conditional independence $Y \perp I_X \mid X, S$ is shown in (a) when $X$ is given and is represented by black downward triangles (▼). These CI patterns for (a) are summarized using the corresponding symbols displayed below the causal graph.
>
> ---
>
> **Q2:** **"The paper does not compare with some existing work with a very similar setting (such as Dai et al. 2025)."**
>
> Thank you for pointing out this related work. Following your suggestion, we have added a dedicated discussion in the revised manuscript. In short, while the two methods appear similar at a high level, the key assumptions and problem formulations are in fact quite different. Specifically,
>
>
> **Assumptions of CDIS (Dai et al.):**
>
> **CDIS** models selection as a one-time process that occurs **only before intervention** and becomes inactive afterward, using an interventional twin graph. Under this assumption, the distribution satisfies $P(Y \mid \text{do}(X) = x, S) = P(Y \mid S)$, meaning that after an intervention on $X$ is applied, the selection variable S no longer influences $Y$.
>
> **Assumptions of GISL:**
>
> In contrast, our work focuses on **biological constraints in genes**—a form of inherent selection bias that exists **both before and after intervention**. These constraints come from basic biological rules, such as essential gene functions or conditions needed for a cell to survive, which do not disappear even when a gene is perturbed. As a result, selection continues to affect the system even after $\text{do}(X)$, violating the CDIS assumption. In our setting, the correct relation is $P(Y \mid \text{do}(X) = x, S) \neq P(Y \mid S)$. **This fundamental difference means that CDIS cannot account for such persistent selection effects, leading to poor performance in settings where biological constraints are present.**
>
> Moreover, in light of your suggestion, we have conducted additional experimental comparing GISL and CDIS (Dai et al) on synthetic datasets (number of variables $|\mathcal{X}|$=15, sample size n = 1500, soft intervention). The results are as below:
>
> | Method | F1| Precision |Recall| SHD | Precision of selection bias|
> |----------|----------|----------|----------|----------|----------|
> | GISL | 80.0 $\pm$ 1.7  | 83.9 $\pm$ 3.2  | 74.1 $\pm$ 1.6 | 4.1 $\pm$ 5.7| 80.8 $\pm$ 11.3|
> | CDIS   |  63.8 $\pm$ 4.5     |  61.4 $\pm$ 4.7  |  68.1 $\pm$ 5.3  |   8.4 $\pm$ 6.4   | 34.1 $\pm$ 9.1 |
>
> We demonstrate that, within the context of specific biological constraints, our method achieves superior performance due to the correct identification results.
>
> ---
> **Q3:**  **The problem setting of the selection variable is vague. The interpretation of the selection variable and perturbation indicator is unclear.**
>
> Thanks for your feedback, which has certainly improved the readability of our paper.
>
> **Problem Setting:**
>
> - **Goal:** Identify regulatory relationships, latent confounders, and selection bias (biological constraints) in gene expression data.
>
> -   **Assumption:** Gene expression is affected by latent confounders and persistent selection bias, as supported by biological evidence (see Lines 29–31 and 33–39).
>
> -   **Solution:** We formulate this as a causal discovery problem under latent confounding and selection bias. By leveraging both observational and gene perturbation (interventional) data, GISL identifies regulatory relationships, latent confounders, and selection bias by capturing differences in **perturbation symmetry** and **intervention effects** (see Section 3.1).
>
> **Clarification on the Selection Variable and Perturbation Indicator:**
>
> As described in Lines 85–88, **the selection variable S imposes a constraint where only samples with S = 1 are observed; data with S = 0 is unobservable.** This formulation is formally discussed in the "M Bias" section on page 5 in [1] and has been widely adopted in subsequent works [2–5]. Thus, our analysis is restricted to the selected subpopulation.
>
> Importantly, **selection bias and intervention are independent**—selection can affect both observational and interventional data. The intervention is designed and classified in Sections 4 and 5 (Pages 5-7) in [6].  The  GISL is designed to **identify** selection bias, not eliminate it. To this end, we introduce a perturbation indicator into an **augmented causal graph**, which enables joint modeling of observational and interventional distributions. This allows CI tests to capture both **perturbation symmetry** and **perturbation effects**, facilitating the identification of regulatory relationships, latent confounders, and selection bias.
>
> > [1] Quantifying Biases in Causal Models: Classical Confounding vs Collider-Stratification Bias. Epidemiology, 2003.
>
> > [2] Causal discovery from data in the presence of selection bias. PMLR, 1995.
>
> > [3] Structural approach to selection bias. Epidemiology, 2004.
>
> > [4] Sample selection bias as a specification error. Econometrica: Journal of the Econometric Society, 1979.
>
> > [5] Endogenous Selection Bias: The Problem of Conditioning on a Collider Variable. Annual Review of Sociology, 2014.
>
> > [6] Variables of causal intervention. UAI, 2004.
>
> ---
> **Q4:**  **"What are the terms like $[I_i, X_j \mid S]$? Are those indicator variables for the conditional independence?"**
>
> Thanks for pointing this out. $[I_i, X_j \mid S]$ denotes that we need to capture the CI relation between the intervention indicator $I_i$ and the variable $X_j$, given the selection variable $S$.
>
> ---
> **Q5:**  **"How scalable is the proposed method? How sensitive is the proposed method to the size of datasets?"**
>
> Scalability is a key consideration in the design of GISL. GISL is built to handle **large-scale gene expression datasets**, as demonstrated in our real-world experiments (Section 4.3 and Appendix F.1, Lines 760–786), where GISL was applied to datasets with over **5,000 observed genes (variables)** and more than **100 perturbed genes**. In comparison, the original nonparametric CI test KCI [8] was only evaluated on graphs with **5 variables**, and baseline methods such as [7] consider at most **20 variables**. These comparisons highlight the practical scalability of GISL beyond prior work.
>
> In terms of sensitivity to dataset size, our experiments on synthetic data (10–25 variables, 500–2,000 samples) and real data show that GISL maintains stable performance across varying sample sizes. This is enabled by two design choices: (1) the use of **skeleton discovery** via FGES, and (2) the application of **pairwise CI tests** based on limited conditioning sets, both of which help ensure computational efficiency as dataset size grows.
>
> > [7] Permutation-Based Causal Structure Learning with Unknown Intervention Targets. UAI, 2020.
>
> > [8] Kernel-based Conditional Independence Test and Application in Causal Discovery. UAI, 2011.
>
> ---
> **Q6:** **"The option in Algorithm 1  $\mathcal{J}==(c)$  and $\mathcal{J}==(d)$  seems to be flipped and inconsistent with the caption of Figure 3."**
>
> Thanks for pointing this out. The options for $\mathcal{J}==(c)$  and $\mathcal{J}==(d)$ in Algorithm 1 were inadvertently flipped. We have corrected this error to ensure consistency between the algorithm and the figure.
>
> ---
> **Q7:**  **"Do the terms regulatory relations and causal relations mean the same thing in this context? The paper seems to use these terms interchangeably."**
>
> Yes, in this context, _regulatory relation_ and _causal relation_ refer to the same concept. We primarily use the term _regulatory relationship_ to align with the gene regulatory network inference (GRNI) context, while _causal relationship_ is used when introducing the formal model to remain consistent with standard causal discovery terminology. This interchangeable use is clarified in Lines 79–81 and is consistent with the GRNI definition in Line 22, where regulatory interactions are modeled as causal effects.
>
> ---
>  If you have any questions, please do not hesitate to let us know. We would greatly appreciate your feedback and are happy to discuss further.

---

> > ### Comment · Reviewer_7UVK · 2025-08-07
> >
> > Thank you for the detailed comments. I now have a better understanding of Figure 3, and many of my previous concerns have been addressed. I believe the current version of the paper could be enhanced through improved structure and clarity. Considering the additional experiments and clarifications provided by the authors, I will increase my score accordingly.

---

> ### Author Response · Authors · 2025-08-04
> **Could you please kindly let us know if our response answers your questions?**
>
> Dear Reviewer 7UVK,
>
> Thank you very much for your time and effort in reviewing our manuscript. **We have incorporated your suggestions to clarify the problem setting and the Figures, and demonstrate GISL’s scalability.**
>
> As the discussion deadline approaches, please let us know if you have any further concerns, and we will be happy to discuss further.
>
> **Kindly note that the OpenReview system may occasionally fail to render LaTeX correctly. If you encounter any unrendered LaTeX, please refresh the page. Thank you.**
>
> Yours sincerely,
>
> Authors of Submission 17977

---

> ### Author Response · Authors · 2025-08-07
> **Could you kindly let us know if there are any remaining concerns we can address?**
>
> Dear Reviewer 7UVK,
>
> Thank you very much for your valuable feedback, which has greatly contributed to improving the clarity and readability of our paper.
>
> As the discussion deadline approaches (with only 48 hours remaining), we would be grateful for the opportunity to address any remaining concerns you may have. If there are any outstanding questions or points that require further clarification, we would be more than happy to respond.
>
>
> With sincere appreciation,
>
> Authors of Submission 17977

---

> ### Author Response · Authors · 2025-08-07
>
> Thank you very much for your encouragement. We are pleased that many of your concerns have been addressed and that Figure 3 is now clearer. We sincerely appreciate your recognition of the additional experiments and revisions, and we are grateful for your willingness to improve the score. Your feedback has been invaluable in strengthening the paper.

---

### Official Review · Reviewer_yzJf · 2025-07-03

**Clarity:** 4
**Significance:** 3
**Originality:** 4
**Rating:** 5
**Confidence:** 4

**Summary:**

This paper introduces GISL, a nonparametric causal discovery algorithm tailored for gene regulatory network inference (GRNI) in the presence of both selection bias and latent confounders. The main innovation hinges on the key insight that *selection bias* can create spurious dependencies in gene expression data, particularly in contexts where only cells meeting certain survival thresholds might be observed in lab settings. GISL is a perturbation-inspired framework to distinguish selection-induced dependencies from true regulation or confounding. GISL is operationalized as a flexible causal framework that leverages conditional independence to differentiate between regulatory, confounded, and selection-based relationships, with a well formulated algorithm (Algo 1). The method is validated on both synthetic and real single-cell gene expression datasets (scRNA-seq perturbation datasets), and is compared to competing methodologies.

**Questions:**

The three main concerns I have, which I believe could improve the paper, are:

**Robustness of Simulation Experiments:**
The simulation study could benefit from a more thorough sensitivity analysis. Methods like GISL are often highly sensitive to both noise levels and the dimensionality of the system—particularly in high-gene-count settings typical in gene expression studies. However, the current simulations appear to focus only on relatively small systems (10–25 variables). It's unclear whether GISL's performance generalizes to more realistic, higher-dimensional scenarios, or more complicated functional forms of the SCM. A systematic evaluation of performance across varying noise levels, SCM complexities, and number of genes would help validate the method’s robustness.

**Realism of the Synthetic Data Generator:**
While SCMs offer a clean framework for simulation, it's not entirely clear how well they reflect the complexities of real gene expression dynamics. More biologically grounded simulators—such as the one introduced in Bhuva et al. (“Differential co-expression-based detection of conditional relationships...”)—could provide a more realistic benchmark. If this is too difficult, then at least some context or justification for using SCMs over existing biological simulators would be helpful, especially if the goal is to evaluate performance in settings that mirror actual gene regulation.

**Need for More Real-World Evidence:**
Given the strong real-world motivation throughout the paper—for instance, the AURKA–TOR1AIP1 example in the introduction—there’s definitely room to further ground the method in real world biological data. One way to improve this would be to shift some theoretical content to the appendix and expand the real data application. Specifically, the paper could benefit from showcasing concrete examples of relationships that GISL correctly flags as arising from selection bias in leukemia and lung cancer—ideally, in pathways with known biological validation. On the flip side, it would also be valuable to highlight where GISL fails or gives ambiguous results. Both successes and limitations will help biologists understand the tangible benefits of GISL.

**Ethical Concerns:**

["NO or VERY MINOR ethics concerns only"]

**Final Justification:**

Following the authors' rebuttal and in light of the additional experiments and clarifications, I have revised my evaluation. The authors have addressed my primary concerns by demonstrating the robustness of GISL across multiple dimensions, including noise levels, structural complexity, and gene counts. These results provide stronger evidence for the generalizability of the method. Furthermore, the inclusion of real-world biological evidence enhances the relevance and interpretability of the approach, particularly for biologically focused audiences.

**Limitations:**

yes

**Paper Formatting Concerns:**

well written and formatted. the problem is well contextualized.

**Quality:**

3

**Strengths And Weaknesses:**

Strengths:
- problem is well motivated and biologically meaningful
- well formulated (and clearly explained) algorithm
- interesting simulation experiments
- strong biological application

Weaknesses (elaborated below):
- simulation experiments can be made more robust
- need more real world evidence

NOTE: authors performed extensive additional experiments to address my concerns listed above.

---

> ### Author Rebuttal · Authors · 2025-07-31
>
> We first sincerely appreciate your insightful suggestions and valuable feedback, which have enabled us to strengthen our method through a more comprehensive evaluation. We have carefully considered the concerns you raised and provide our detailed responses below.
>
> ---
> **Q1:** **Robustness of Simulation Experiments.**
>
> In light of your suggestion, we have conducted comprehensive experiments to evaluate the robustness of GISL as follows:
>
> **Noise level**
>
> We conducted experiments to evaluate the robustness of GISL under varying levels of Gaussian noise variance, where the noise mean was randomly sampled from a uniform distribution U(0, 4). **Experimental results demonstrate that GISL’s performance improves as Gaussian noise variance increases, because larger distributional changes are more easily detected by conditional independence (CI) tests, yielding more accurate inference.**
>
> Experimental setting: hard intervention, 15 variables, 1500 samples.
> | Variance of noise | F1| Precision |Recall| SHD |
> |----------|----------|----------|----------|----------|
> | U(0,2) | 69.6 $\pm$ 1.9  | 80.8 $\pm$ 4.3  | 61.8 $\pm$ 1.9 | 5.5 $\pm$ 6.5|
> | U(0,4)    |  72.6 $\pm$ 1.8     |  93.5 $\pm$ 1.7  |  62.1 $\pm$ 3.2  |   4.7 $\pm$ 4.6   |
> | U(2,4)    |  77.4 $\pm$ 0.8     |  89.7 $\pm$ 1.4  |  69.2 $\pm$ 1.1  |   4.3 $\pm$ 3.8   |
>
> Experimental setting: hard intervention, 15 variables, 500 samples.
>
> | Variance of noise | F1| Precision |Recall| SHD |
> |----------|----------|----------|----------|----------|
> | U(0,2)    |  61.6 $\pm$ 0.9  |  86.8 $\pm$ 0.2  |  50.1 $\pm$ 1.7  |   5.8 $\pm$ 1.4   |
> | U(0,4) | 67.4 $\pm$ 1.8  | 87.2 $\pm$ 2.9  | 56.1 $\pm$ 1.7 | 5.7 $\pm$ 6.6|
> | U(2,4)    |  67.9 $\pm$ 1.0     |  88.8 $\pm$ 1.3  |  56.1 $\pm$ 1.5  |   5.6 $\pm$ 4.0   |
>
> **SCM complexity**
>
> We conducted experiments to evaluate the robustness of GISL under varying levels of structural complexity, quantified by the number of edges in the graph. **Experimental results indicate that GISL’s performance experiences a slight decline as graph structural complexity increases.**
>
> Experimental setting: hard intervention, 15 variables, 1500 samples.
> | Number of edges | F1| Precision |Recall| SHD |
> |----------|----------|----------|----------|----------|
> | 15 | 69.6 $\pm$ 1.9  | 80.8 $\pm$ 4.3  | 61.8 $\pm$ 1.9 | 5.5 $\pm$ 6.5|
> | 20    |  67.6 $\pm$ 0.6     |  92.8 $\pm$ 0.9  |  59.4 $\pm$ 0.9  |   7.2 $\pm$ 3.3   |
>
> Experimental setting: hard intervention, 15 variables, 500 samples.
>
> | Number of edges | F1| Precision |Recall| SHD |
> |----------|----------|----------|----------|----------|
> | 15    |  61.6 $\pm$ 0.9  |  86.8 $\pm$ 0.2  |  50.1 $\pm$ 1.7  |   5.8 $\pm$ 1.4   |
> | 20 | 59.2 $\pm$ 1.3  | 90.1 $\pm$ 1.2  | 49.4 $\pm$ 1.5 | 8.8 $\pm$ 4.7|
>
> **The number of genes**
>
> We conducted experiments to evaluate the robustness of GISL under a varying number of genes. **Experimental results indicate that GISL’s performance remains stable and even improves as the gene count increases, owing to the greater sparsity of the corresponding graphs.**
>
> Experimental setting: hard intervention, 1500 samples.
> | Number of genes | F1| Precision |Recall| SHD |
> |----------|----------|----------|----------|----------|
> | 15 | 69.6 $\pm$ 1.9  | 80.8 $\pm$ 4.3  | 61.8 $\pm$ 1.9 | 5.5 $\pm$ 6.5|
> | 25    |  67.3 $\pm$ 0.3     |  85.8 $\pm$ 1.2  |  56.1 $\pm$ 0.9  |   10.8 $\pm$ 1.9   |
> | 50    |  71.2 $\pm$ 1.6     |  90.7 $\pm$ 1.3  |  62.4 $\pm$ 2.1  |   14.2 $\pm$ 6.8   |
>
> Experimental setting: hard intervention, 15 variables, 500 samples.
>
> | Number of edges | F1| Precision |Recall| SHD |
> |----------|----------|----------|----------|----------|
> | 15    |  61.6 $\pm$ 0.9  |  86.8 $\pm$ 0.2  |  50.1 $\pm$ 1.7  |   5.8 $\pm$ 1.4   |
> | 25 | 64.3 $\pm$ 0.6  | 90.8 $\pm$ 0.2  | 50.6 $\pm$ 1.1 | 11.2 $\pm$ 3.6|
> | 50 | 68.8 $\pm$ 1.4  | 87.2 $\pm$ 0.5  | 53.7 $\pm$ 0.9 | 16.3 $\pm$ 7.6|
>
> For larger datasets, please see Figures 11 and 13, which present real-world experiments on gene expression data encompassing over **5,000 observed genes** and **more than 100 perturbed genes**.
>
> ---
> **Q2:** **Realism of the Synthetic Data Generator.**
>
> In light of your suggestions, we consider biological simulators, which simulate the regulatory relationship by using the normalised-Hill differential equations and gene expression is a truncated distribution. Experimental results are as follows:
>
> Experimental setting: hard intervention, 15 variables, 1500 samples.
> | Regulatory function | F1| Precision |Recall| SHD |
> |----------|----------|----------|----------|----------|
> | Random selected nonlinear | 69.6 $\pm$ 1.9  | 80.8 $\pm$ 4.3  | 61.8 $\pm$ 1.9 | 5.5 $\pm$ 6.5|
> |  Normalised-Hill differential equations   |  67.6 $\pm$ 1.7     |  83.1 $\pm$ 2.6  |  59.7 $\pm$ 3.2  |   5.9 $\pm$ 5.3   |
>
>
> Experimental setting: hard intervention, 15 variables, 500 samples.
>
> | Regulatory function | F1| Precision |Recall| SHD |
> |----------|----------|----------|----------|----------|
> | Random selected nonlinear   |  61.6 $\pm$ 0.9  |  86.8 $\pm$ 0.2  |  50.1 $\pm$ 1.7  |   5.8 $\pm$ 1.4   |
> | Normalised-Hill differential equations | 59.4 $\pm$ 1.3  | 85.7 $\pm$ 2.3  | 48.9 $\pm$ 2.1 | 6.4 $\pm$ 3.6|
>
> Experimental results demonstrate stable performance on data generated by SCM-driven biological simulators.
>
> ---
> **Q3:** **Need for More Real-World Evidence.**
>
>  According to your suggestion, we have added more real-world evidence to support the results of GISL as follows:
>
> **Human Lung Epithelial Cells**
>
> GISL reports that there exist biological constraints between gene CNN1 and CDKN1A. **CNN1** (Calponin 1) is normally low in healthy lung epithelium but markedly increases during epithelial–mesenchymal transition or fibrotic remodeling. [1] **CDKN1A** (_p21_) is a cell-cycle inhibitor that surges under DNA damage or stress (e.g., p53 activation), causing growth arrest or senescence; indeed, lung injury models and fibrotic lungs (IPF) show abnormally high p21 in alveolar epithelial cells [2]. These observations imply that when lung epithelial cells are pushed toward a damaged or transitioning state, both CNN1 and p21 tend to rise, which suggests that the cell may need to keep its combined activity in check to maintain normal function. Excessive co-elevation of CNN1 (indicating mesenchymal/fibrotic shift) and CDKN1A (indicating cell-cycle arrest) could drive cells into an unsustainable state (senescence or fibrosis), so a balance in their expression is likely required to preserve viability and tissue homeostasis. This suggests a form of _biological constraint_ whereby lung epithelial cells limit the concurrent upregulation of CNN1 and CDKN1A to avoid tipping into pathological remodeling or loss of proliferative capacity.
>
> **K562 Lukemia Cells**
>
> GISL reports that there exist biological constraints between gene ELF1 and gene GABPA. **ELF1**, an ETS-family transcription factor, regulates hematopoietic differentiation and immune genes; in K562 cells, it controls the G₁/S transition via CDKN1A (p21), with overexpression inducing apoptosis and underexpression promoting unchecked proliferation [3]. **GABPA**, a nuclear respiratory factor, orchestrates mitochondrial biogenesis and metabolism through the PI3K/Akt/mTOR axis; its depletion triggers G₀/G₁ arrest, and its binding affinity (pKD) and phosphorylation response (pEC₅₀) underscore a requirement for dosage matching downstream signaling intensity [4]. Both factors converge on stress and apoptotic pathways—GABPA targets (Caspase-9, Bcl-2) overlap with ELF1-regulated DNA-damage genes—and respond coordinately to PI3K/Akt perturbations (e.g., LY294002) and autophagy induction (e.g., in K562/ADM cells), suggesting K562 cells impose a constraint on ELF1 and GABPA expression to maintain proliferative homeostasis and survival [5,6].
>
> > [1] Alveolar epithelial cells express mesenchymal proteins in patients with idiopathic pulmonary fibrosis. American Journal of Physiology-Lung Cellular and Molecular Physiology 2011.
>
> > [2] P21Waf1/Cip1/Sdi1 and p53 expression in association with DNA strand breaks in idiopathic pulmonary fibrosis. American journal of respiratory and critical care medicine 1996.
>
> > [3] SIRT1 is a critical regulator of K562 cell growth, survival, and differentiation. Expression Cell Research 2016.
>
> > [4] Decrypting drug actions and protein modifications by dose-and time-resolved proteomics. Science  2023.
>
> > [5] A genetic map of the response to DNA damage in human cells, Cell  2020.
>
> > [6] Elf1 promotes transcription-coupled repair in yeast by using its C-terminal domain to bind TFIIH. Nature Communication 2024.

---

> ### Author Response · Authors · 2025-08-04
> **Could you please let us know whether our responses properly addressed your concern?**
>
> Dear Reviewer yzJf,
>
> Thank you for taking the time to review our paper and for your helpful suggestions. Your feedback on evaluating the robustness of GISL and real-world extension was constructive in improving our paper. We hope that our responses have addressed your concerns.
>
> Due to the limited time for rebuttal discussion, we look forward to receiving your feedback at your earliest convenience and the opportunity to respond to it.
>
> **Kindly note that the OpenReview system may occasionally be unable to compile LaTeX correctly. If you encounter any uncompiled equations, please try refreshing the page. Thank you for your patience.**
>
> Yours sincerely,
>
> Authors of Submission 17977

---

> > ### Comment · Reviewer_yzJf · 2025-08-05
> > **Response to authors**
> >
> > Thank you for your thorough and thoughtful responses to my comments.
> >
> > I sincerely appreciate the effort you’ve put into addressing the concerns in such a short time. It’s particularly reassuring to see the robustness of GISL demonstrated across varying noise levels, structural complexities, gene counts, and ground truth dynamics (Hill equations). The addition of real-world biological evidence—especially the examples involving lung epithelial and K562 leukemia cells—adds valuable context and will certainly resonate with a biologically inclined audience.
> >
> > Based on these improvements, I have updated my score accordingly.

---

> > > ### Author Response · Authors · 2025-08-06
> > >
> > > We sincerely appreciate your thoughtful and constructive feedback. Your comments have helped strengthen our work by encouraging a more comprehensive evaluation and providing valuable recognition of our contributions. Thank you again for your time and support.

---

### Official Review · Reviewer_TNYS · 2025-07-03

**Clarity:** 2
**Significance:** 2
**Originality:** 2
**Rating:** 5
**Confidence:** 4

**Summary:**

This paper is concerned with inferring latent confounders and selection bias in gene regulatory networks with interventional data. The paper shows that interventional data implies different d-separations in the graphical models for direct cause, existence of latent confounder, and existence of selection bias. Next, an algorithm, GISL, is introduced that uses both observational and interventional data to infer the causal graph as well as the existence of confounders and selection bias. The author's test this method in synthetic settings, where they show that it can identify selection bias. They then test the method on synthetic graphs that have selection bias, and on three gene perturbation datasets.

**Questions:**

- Why are the experiments only done with selection bias and not with latent confounders as well? Most of the paper talks about identifiability of both.
- It is hard to interpret the results of 4.1 without a proper baseline. Maybe a baseline where the selection criteria is observed (so it reduces to finding a collider) may provide an upper bound on the performance?
- There is no mention of the actual conditional independence test used or other experimental details in the Appendix.

**Ethical Concerns:**

["NO or VERY MINOR ethics concerns only"]

**Final Justification:**

Though the presentation of the work can be improved (which is partly addressed by the authors), I do believe taking selection bias into account is an overlooked area.

Although I agree with the concerns raised by yzJf, they have been partly addressed in my and their responses. I also think these concerns are generally valid for almost all causal discovery methods.

**Limitations:**

Yes

**Quality:**

2

**Strengths And Weaknesses:**

**Strengths:**
- The idea of the paper is simple yet effective.

**Weaknesses:**
- I'm unsure why the paper explicitly looks at gene regulatory networks when the method is more generally for causal discovery with interventional data. I believe this may reduce the perceived significance of the work.
- The related work should be in the main paper, and not in the appendix. It should also be emphasised in the related work whether methods require only observational data, or also interventional data (like the current work). A lot of the criticisms of related work, that they require functional assumptions is because they do not require interventional data. The related work could also be more complete. The novelty and differences of this work in comparison to other works is not made very clear.
- Some of the claims made in the paper could be more accurate. It seems that the algorithm only allows for identifying latent confounders with interventions on all variables (this statement is missing in Theorem 3.7). Even in this case, there are certain cases that are not identifiable (like figure 7(d) as mentioned in Appendix D).
- The claims about unobserved confounder are not backed by experiments (see question).
- The SID (structural intervention distance) metric might also be a more natural measure of how good the recovered graph is (as opposed to just SHD).


**Comments:**
- The figures need to be bigger to be legible along with the rest of the text. Figure 2 is not decipherable at all.
- Figure 3 is quite hard to understand and it took me a while to figure out what the shapes were. I'm also fairly sure 3 (b) top is incorrect.
- L234: Where is example 3.2? Do you mean example 2?
- L271: It is claimed that you are using nonparametric SCMs for complexity, but the functions chosen are actually quite simple. Would functions drawn from a GP or an NN not be more appropriate?

---

> ### Author Rebuttal · Authors · 2025-07-31
>
> We sincerely thank you for the time dedicated to reviewing our paper and valuable feedback. Please find the response to your comments and questions below.
>
> ---
> **Q1:** **Why does the paper explicitly look at gene regulatory networks when the method is more generally for causal discovery with interventional data?**
>
> Thank you for this insightful suggestion. Indeed, our method is broadly applicable to general causal discovery, but we chose the gene regulatory network inference (GRNI) framework because it was inspired by a real‐world challenge—the pervasive yet often overlooked selection (survival) bias in biological data—and directly addressing this issue offers immediate value for GRNI and beyond.
>
> Specifically, we focus on GRNI because:
> 1. GRN inference is a classical causal discovery task with strong biological significance, supported by well-established gene perturbation technologies that provide abundant interventional data.
> 2.  Despite being a classical problem, modeling gene expression from a causal perspective remains challenging due to latent confounding and overlooked selection bias, making it a meaningful testbed for our method.
>
> ---
> **Q2:** **Related works should be in the main paper and could be more complete. The difference between GISL and related works should be clear.**
>
> In the revised version, **we will move the discussion of related work into the main paper and expand it accordingly.** Due to limited space, we list representative causal discovery methods for comparison in settings and highlight the novelty and difference compared with related works as follows:
>
> | Mehtod | Require Interventional data|Allow latent confounder |Allow selection bias | nonparametric assumption |
> |----------|----------|----------|----------|----------|
> | [1] |✅| ❌  | ❌ | ✅|
> | [2] | ❌ | ✅| ✅|✅|
> | [3] | ❌ | ✅| ❌| ❌|
> | [4] | ✅ | ✅|❌|✅ |
> | [5] | ❌ |❌|  ✅|❌|
> | Ours    |✅|✅|✅|✅|
>
> In comparison, **GISL is novel in utilizing interventional data to identify relationships, latent confounders, and selection bias in a general setting without relying on parametric assumptions**, and is specifically tailored to handle the **biological selection constraints** that arise in gene regulatory networks.
>
> > [1]. Permutation-Based Causal Structure Learning with Unknown Intervention Targets. UAI 2020.
>
> > [2]. Iterative causal discovery in the possible presence of latent confounders and selection bias. NeurIPS 2021.
>
> > [3]. A Versatile Causal Discovery Framework to Allow Causally-Related Hidden Variables. ICLR 2023.
>
> > [4]. Characterization and Learning of Causal Graphs with Latent Variables from Soft Interventions. NeurIPS 2019.
>
> > [5]. Identifying selection bias from observational data. AAAI 2018.
>
> ---
> **Q3:** **The statements about the identifiability in Theorem 3.7 could be more accurate. There are certain cases that are not identifiable (like figure 7(d) as mentioned in Appendix D).**
>
> We agree with you. We have revised the statement in Theorem 3.7 to make it more accurate as follows:
>
> The causal relationships, selection processes, and latent confounders are identifiable up to the equivalence classes of four types of CI patterns in Figure 3 **among the variables that are subject to intervention.** The presence of selection and latent confounders (existing or not) is identifiable among these variables.
>
> Additionally, as shown in Figure 7(d) in Appendix D, **certain structures such as $X \rightarrow L \leftarrow Y$ differ in causal structure but yield the same Conditional Independence (CI) patterns, placing them in the same Markov equivalence class under PAG representation**. While the exact structure may not be distinguishable, **the presence of latent confounders between X and Y can still be inferred**. Thus, GISL can reliably identify whether selection bias or latent confounders exist among the intervened variables and identify causal structures up to the equivalence class.
>
> ---
> **Q4:** **Lacking the accuracy of identifying latent confounders.**
>
>  We report GISL’s ability to identify latent confounders on **synthetic datasets** with 1500 samples as follows:
>
> | # Variables| 10| 15 |20| 25|
> |----------|----------|----------|----------|----------|
> | hard| 62.5 $\pm$ 22.8  | 63.2 $\pm$ 20.6  | 68.8 $\pm$ 15.2  | 66.7 $\pm$ 14.7 |
> | soft  |   66.8 $\pm$ 25.3   |   68.4 $\pm$ 22.9   | 70.4 $\pm$ 18.5    |    71.2 $\pm$ 15.6  |
>
> We focus on selection bias in our experiments for the following reasons:
>
> 1.  **One of the main contributions of this work is to highlight the often-overlooked role of selection bias in gene expression data**, and to propose GISL specifically to identify it.
> 2.  **Selection bias can be systematically evaluated** on both synthetic and real-world datasets, as the selection processes (e.g., biological constraints) are either known or can be reasonably assumed. In contrast, **latent confounders can only be evaluated on synthetic datasets**, since there is no ground truth or systematic benchmark for latent confounding in real-world gene expression data.
>
> ---
> **Q5:** **The SID (structural intervention distance) metric might also be a more natural measure of how good the recovered graph is.**
>
> Thank you for the suggestion. GISL focuses on recovering the **graphical structure**, and **SHD**, which directly measures structural differences globally, is a commonly used and standard evaluation metric in this context. While **SID** provides a meaningful assessment of **local interventional reasoning**, it focuses on **ancestral relationships** and is generally more tolerant of structural discrepancies. As a result, SID may overlook structural inaccuracies that do not impact interventional distributions.
>
> Nevertheless, we appreciate your comment and are happy to incorporate it for a more comprehensive analysis. According to your suggestion, we evaluate the SID under varying sample size n for hard intervention as follows:
> | # Variables| 10| 15 |20| 25|
> |----------|----------|----------|----------|----------|
> | n= 500| 4.5 $\pm$ 1.5  | 5.1 $\pm$ 1.2  | 5.8 $\pm$ 0.9  | 7.7 $\pm$ 1.0 |
> | n=1500  |   4.3 $\pm$ 1.2   |    4.8 $\pm$ 1.9   | 5.6 $\pm$ 1.1     |   7.3 $\pm$ 1.3  |
> | n=2000  |    4.2 $\pm$ 1.1  |    4.7 $\pm$ 1.4   | 5.8 $\pm$ 1.0     |    7.1 $\pm$ 0.9  |
>
> ---
> **Q6:** **Figure 2 is not decipherable.**
>
> We will revise and reformat Figure 2 to improve its clarity and readability in the final version of the paper.
>
> ---
> **Q7:** **Figure 3 is quite hard to understand, and (b) top is incorrect.**
>
> We thank you for your patience in interpreting Figure 3. We have added a detailed description for better clarity and corrected the mistake in panel (b). Additionally, we have added a detailed explanation to help readers interpret the figure more easily.
>
> Specifically, Figure 3 (f) serves as a **reference table** summarizing the CI patterns for each target gene pair. In this table, **black symbols** indicate that the corresponding CI relation **holds** (i.e., the variables are conditionally independent, while **white symbols** indicate that the CI relation **does not hold** (i.e., the variables are conditionally dependent).
>
> **Taking Figure 3(a) as an example,**  the structure exhibits conditional dependence $Y \not\perp I_X \mid S$, which is represented by the white upward triangles (△) in the reference table. The conditional independence $Y \perp I_X \mid X, S$ is shown in (a) when $X$ is given and is represented by black downward triangles (▼). These CI patterns for (a) are summarized using the corresponding symbols displayed below the causal graph.
>
> ---
> **Q8:**  **Incorrect reference to example 2.**
>
> Thanks for pointing this out. We have corrected it.
>
> ---
> **Q9:** **L271: The function in nonparametric simulation is simple. Would functions drawn from a GP or an NN not be more appropriate?**
>
> According to your suggestion, we have expanded the function pool to include a **neural network (NN)** with a hidden dimension of [5, 5] and ReLU activation. The regulatory function is now randomly sampled from a set that includes linear, quadratic, sin, log, and NN functions. The results of number of variables $\mathcal{X}$ = 15, number of samples n = 1500  for soft intervention are as follows:
>
> | Method | F1| Precision |Recall| SHD | SID|
> |----------|----------|----------|----------|----------|----------|
> | Without NN | 80.0 $\pm$ 1.7  | 83.9 $\pm$ 3.2  | 74.1 $\pm$ 1.6 | 4.1 $\pm$ 5.7| 3.6 $\pm$ 1.2|
> | With NN    |  77.1 $\pm$ 1.5     |  82.4 $\pm$ 2.5  |  73.1 $\pm$ 1.6  |   4.4 $\pm$ 6.4   | 4.1 $\pm$ 1.1 |
>
> ---
> **Q10:** **Baselines for identifying colliders as upper bound performance.**
>
> Following your advice, we conducted experiments on synthetic data comprising 15 variables and 2000 samples under soft interventions to evaluate the identifiability of GISL in detecting colliders, as detailed below:
>
> | # | 1 | 2 | 3 | 4 |
> |----------|----------|----------|----------|----------|
> | Selection process | 78.5 $\pm$ 15.6  | 76.1 $\pm$ 14.5  | 70.6 $\pm$ 17.7 | 69.5 $\pm$ 14.3|
> | Collider    |  90.0 $\pm$ 5.0     |  85.0 $\pm$ 5.3  |  83.3 $\pm$ 5.6  |   82.5 $\pm$ 3.9   |
>
> ---
> **Q11:** **Lacking the actual conditional independence test used or other experimental details.**
>
>  We have added detailed descriptions of the experimental setup as follows:
> Specifically, following Algorithm 1, the first step—**skeleton discovery**—is performed using **FGES** with the **BIC score** to identify the conditional dependence. In Step 3, we capture CI patterns from both observational and interventional data. For the **nonparametric CI testing**, we use the **Kernel-based Conditional Independence (KCI)** test.
>
> ---
> If any aspect remains unclear, please do not hesitate to let us know; we would be more than happy to discuss it further with you.

---

> > ### Comment · Reviewer_TNYS · 2025-08-05
> > **Response to authors**
> >
> > > SID may overlook structural inaccuracies that do not impact interventional distributions.
> >
> > I am unsure about this statement as SID counts the number of falsely inferred interventional distances.
> >
> > I am happy to raise my score as my other concerns have been addressed.

---

> ### Author Response · Authors · 2025-08-04
> **Could you please let us know whether our responses properly addressed your concern?**
>
> Dear Reviewer TNYS,
>
> Thank you again for your valuable time dedicated to reviewing our paper and for your helpful suggestions. We particularly appreciate your questions regarding the complexity of the simulation, theorem, and related works. So we are eager to see whether our responses properly addressed your concerns and would be grateful for your feedback.
>
> **Kindly note that the OpenReview system may occasionally be unable to compile LaTeX correctly. If you encounter any uncompiled equations, please try refreshing the page. Thank you for your patience.**
>
> With best wishes,
>
> Authors of Submission 17977

---

> ### Author Response · Authors · 2025-08-06
>
> We are so happy that your other concerns have been addressed, and we sincerely thank you for your recognition of our work.
>
> Regarding the statement on SID, thank you for pointing this out. What we intended to convey is that SID and SHD capture **complementary** aspects of model evaluation: SHD penalizes all structural inaccuracies, while SID quantifies differences in the intervention distributions (SID counts the number of falsely inferred interventional distances, as you correctly pointed out).  For a more comprehensive evaluation, we will revise the manuscript to clarify this more carefully to avoid any confusion and will put results on both metrics in the updated version.
>
> We sincerely appreciate your thoughtful and constructive feedback, which has helped us present a more comprehensive evaluation. Thank you.

---

### Note · Authors · 2025-08-15

Dear AC and Reviewers

We thank the AC for your time and contribution to the community, and we are grateful to the reviewers for their thoughtful suggestions and kind recognition of our contribution. To facilitate your discussion, we summarize the rebuttal below.

|Concerns & Suggestions|By|Our Response|
| --- | --- | --- |
|Robustness of GISL|Reviewers TNYS, yzJf |We have incorporated comprehensive experiments varying noise levels, edge density (1,1.4), realism of the synthetic data generator, and number of nodes (10~50) to demonstrate the robustness of GISL.|
|Comprehensive evaluation|Reviewers TNYS, yzJf|We have strengthened the evaluation by adding the recommended SID metric and by the real-world biological evidence.|
|Readability of the reference table in Figure 3|Reviewers TNYS, 7UVK, 7xy2|We have updated the Figure 3 caption with a clear description of all symbols in the reference table and included an illustrative example to improve clarity.|
|Baselines| Reviewers TNYS, 7UVK|We have added the upper-bound performance of GISL for collider identification and a comparison with CDIS, placing the results in the experiments section to assess effectiveness.|
| More detailed discussion with related works|Reviewer TNYS|We have moved the related-work discussion into the main text and clarified differences in problem settings relative to existing causal-discovery methods.|
| Conditional independence (CI) test explanation |Reviewer TNYS|We have clarified that our nonparametric CI testing uses the Kernel-based Conditional Independence (KCI).|
|Clarification of technical terminology|Reviewer 7UVK, 7xy2|We have added explicit definitions of commonly used causal terms in the Preliminaries to make the paper more accessible to a broader audience.|
|Moderate recall|Reviewer 7xy2|We have added illustrative examples and a detailed analysis of GISL’s identifiability to explain the observed recall.|

We are pleased that Reviewer TNYS, Reviewer yzJf, and Reviewer 7UVK found our response helpful and that their concerns have been addressed. We also appreciate that Reviewer 7xy2 can better understand the paper with our response, and kindly retained the confidence score (1) to only reflect his/her difficulty in understanding the theoretical foundations of causal discovery and Figure 3.

Sincerely,

The Authors of Submission 17977

---

### Decision · Program_Chairs · 2025-09-17

**Decision:**

Accept (poster)

**Comment:**

The submission develops an algorithm for gene regulatory network inference in the presence of selection bias. Such bias is manifested in the form of symmetric perturbation effects, which cannot be explained by latent confounders assuming a causal DAG. Reviewers overall appreciated the efficacy and strong motivation of the idea. Reviewers raised concerns relating to comparisons with existing work (7UVK, TNYS), technical clarity (7UVK), lack of experiments with both confounders and selection bias (TNYS), lack of sensitivity analysis (yzJf), and lack of real-world case studies (yzJf). In discussions, authors provided additional results with latent confounders, more thorough sensitivity analyses, and  real-world case studies. Reviewers indicated that most of these concerns were addressed to their satisfaction, leading to a consensus towards acceptance.